## PROCEEDINGS A

optics, light microscopy

X-ray optical elements, X-ray microscopy, synchrotron radiation, X-ray free-electron lasers, nonlinear X-ray optics

**Author for correspondence:**
H. N. Chapman
e-mail: henry.chapman@desy.de

An invited Perspective to mark the election of Henry Chapman to the fellowship of the Royal Society in 2020.

# High-resolution achromatic X-ray optical systems for broad-band imaging and for focusing attosecond pulses

H. N. Chapman[1,3,4,5] and S. Bajt[2,3]

[1]Center for Free-Electron Laser Science, [2]Deutsches Elektronen-Synchrotron DESY, Notkestrasse 85, 22607 Hamburg, Germany
[3]The Hamburg Centre for Ultrafast Imaging, and [4]Department of Physics, Universität Hamburg, Luruper Chaussee 149, 22761 Hamburg, Germany
[5]Molecular and Condensed Matter Physics, Department of Physics and Astronomy, Uppsala University, Box 516, 751 20 Uppsala, Sweden

HNC, 0000-0002-4655-1743; SB, 0000-0002-7163-7602

Achromatic focusing systems for hard X-rays are examined which consist of a refractive lens paired with a diffractive lens. Compared with previous analyses, we take into account the behaviour of thick refractive lenses, such as compound refractive lenses and waveguide gradient index refractive lenses, in which both the focal length and the position of the principal planes vary with wavelength. Achromatic systems formed by the combination of such a thick refractive lens with a multilayer Laue lens are found that can operate at a focusing resolution of about 3 nm, over a relative bandwidth of about 1%. With the appropriate distance between the refractive and diffractive lenses, apochromatic systems can also be found, which operate over relative bandwidth greater than 10%. These systems can be used to focus short pulses without distorting them in time by more than several attoseconds. Such systems are suitable for high-flux scanning microscopy and for creating high intensities from attosecond X-ray pulses.

# 1. Introduction

Diffractive optics such as multilayer Laue lenses (MLLs) and sputter-sliced zone plates are currently under development as a means to achieve imaging at 1 nm resolution for X-ray wavelengths of 0.1 nm or less [1,2]. These optical elements are essentially volume holographic optical elements constructed with nanometre layer thicknesses, and lens thicknesses in the direction of beam propagation of tens of micrometres, as needed to achieve efficient beam deflection by Bragg diffraction for high numerical aperture (NA) lenses. Being diffractive optics, such lenses suffer from chromatic aberrations, with a focal length that varies inversely with wavelength, $\lambda$. Each layer in a diffractive optic imparts an additional wave of optical path to create the (first-order) focus by constructive interference. In a lens consisting of $N$ layers, the arrival time of light at the focus will vary by $N\lambda/c$, which may considerably stretch attosecond- or femtosecond-duration X-ray pulses in ultrafast imaging applications. No matter the duration of the illumination, a deviation of the wavelength by $\Delta\lambda$ will result in a cumulative path error of $N\Delta\lambda$ as compared with the design condition. To achieve diffraction-limited performance, this error must be no greater than a wavelength, requiring that $\Delta\lambda/\lambda < 1/N$. Thus, in designing the optical system for an X-ray microscope or to focus ultrashort pulses to obtain high intensities, one may have to choose between limiting the bandwidth of the illumination—resulting in a loss of throughput and a longer pulse duration—or limiting the size of the lens through the number of periods $N$. For example, the spectrum of a typical undulator device may have a relative bandwidth of about 1/300, as given by the number of periods in the undulator. A lens of 300 bi-layer periods to utilize the full spectrum for imaging at a resolution of, say, 1 nm at a wavelength of 0.05 nm would have a radius of only 300 nm and a focal length of 24 $\mu$m. This lens would preserve the transform-limited pulse duration of 50 attoseconds. This calculation is based on the result that the bi-layer periods in a diffractive optic of focal length $f$ are positioned according to $y_N^2 \approx Nf\lambda$, producing a resolution $\delta_r = \lambda/(2NA) = \lambda f/(2y_N) = y_N/(2N)$, so that $y_N = 2N\delta_r$. Expressed another way, the focal length varies inversely with wavelength, giving a dispersion $\Delta f/f = -\Delta\lambda/\lambda$ or a dispersive power of $V = -1$, where

$$V = \left(\frac{\Delta f}{f}\right) \Big/ \left(\frac{\Delta\lambda}{\lambda}\right) = \frac{\lambda}{f}\frac{\partial f}{\partial\lambda}. \tag{1.1}$$

The change in focus $\Delta f$ must remain within the depth of focus of the lens, which itself varies quadratically with the resolution length $\delta_r$.

A similar analysis finds that refractive lenses suffer even harsher limitations. Typical refractive indices of materials in the X-ray regime are slightly less than unity and expressed as $n = 1 - \delta$, with the decrement $\delta$ proportional to $\lambda^2$ at wavelengths away from absorption edges. Given that the focal length of a refractive lens is proportional to $1/(n-1)$ and thus inversely proportional to the square of the X-ray wavelength, the dispersion of a refractive lens in the X-ray regime is given by $V = -2$, which is twice the dispersion experienced by diffractive lenses. A refractive lens therefore stretches a pulse by twice the amount than does a diffractive lens of the same focal length. For a similar resolution and bandwidth, a refractive lens would require an even shorter focal length than considered above to avoid chromatic aberrations.

Regardless of whether we wish to use them in a full-field imaging microscope, to focus a beam to a small spot for a scanning microscope or to focus a short pulse to a small spot to achieve high intensities, the short focal-length lenses of these examples would bring several practical inconveniences. The field of view would be limited to a width that is comparable to the diameter of the lens [3] and the working distance limits the size of objects that can be examined in a tomographic setting. The lens must be positioned near the source or to an image of that source, where the beam size matches the diameter of the lens, placing high demands on the beamline design and optics. Ideally, lenses more than 100 times larger would be preferred, giving focal lengths of millimetres. A diffractive lens like an MLL would then consist of tens of thousands of layers. This increase in $N$ would require a corresponding reduction of tolerable bandwidth to 1/100th of that available and would lead to a stretching of pulses by tens of femtoseconds.

This analysis naturally leads to reflective optics as a basis for a high-throughput or short-pulse imaging system. Mirrors are achromatic and hence can focus radiation of a very broad bandwidth to a small spot, without appreciably stretching the pulse [4]. Kirkpatrick–Baez (KB) mirrors, for example, have been fabricated with resolutions well below 10 nm [5,6], and the use of refractive phase plates to compensate measured aberrations of these systems should further improve their performance [7,8]. The NA of KB optics is limited by the critical angle of reflection but this can be increased using multilayer reflective coatings with some loss of tolerable bandwidth and corresponding increase in pulse response time [9].

Another attractive approach for broad-band or short-pulse applications is to form an achromat by pairing a diffractive lens with a refractive lens. With the appropriate choice of the ratio of focal lengths of the two lenses, the dispersion of the diffractive lens can be compensated by that of the refractive lens and still provide a residual focusing effect [10–13]. An achromat design nulls the linear term in a series expansion of the focal length of the optical system as a function of the relative wavelength deviation $\Delta\lambda/\lambda$, leaving a quadratic dependence so that an equal focal length can be obtained for two distinct wavelengths. This typically provides a bandwidth of 1% or more [11–13], even with diffractive lenses with tens of thousands of layers. Apochromatic designs can also be made. In these, the quadratic term is also brought to zero, which can be achieved with the right choice of separation of the diffractive and refractive lenses [12–14]. In this case, the dominant dependence of the focal length on wavelength is cubic and so an equal focal length can be obtained for three distinct wavelengths, broadening the tolerable bandwidth to 10% or more. This corresponds to a correction of group-velocity dispersion in the lens system, to keep pulse stretching to below about 10 wavelengths or below 2 as for $\lambda = 0.05\,$nm. The refractive–diffractive achromat might offer a cheaper and more compact focusing system than KB mirrors, possibly also at higher resolution.

Here, we examine achromat and apochromat designs to focus short-wavelength X-rays for imaging modalities such as scanning Compton X-ray microscopy [15,16], scanning fluorescence microscopy [3], ptychography [17] and projection imaging [18]. The achievable exposure times of these schemes are usually limited by the available flux that can be focused in a small spot, which could be significantly increased by the ability to accept a larger bandwidth from the source (such as the full width of a harmonic of an undulator device at a modern synchrotron radiation facility). In addition, as attosecond-duration hard X-ray pulses become available at X-ray free-electron lasers [19,20] and compact accelerator sources [21], there is a need to efficiently focus broad-band pulses to create high intensities for nonlinear X-ray optics experiments [22]. These goals demand focused spot sizes considerably smaller than 10 nm over relative bandwidths of several per cent. While the focusing achromat design requires the refractive lens to be diverging (that is, have a negative focal length), both lenses must have comparable power.[1] We present an overview of such systems in §2 and find conditions that give achromatic and apochromatic focusing. Given the dispersions of diffractive and refractive lenses mentioned above, there are two geometries that give achromatic conditions: Type I, consisting of a negative refractive lens followed by a positive diffractive lens, and Type II, where the positive diffractive lens is followed by the negative refractive lens. High-NA MLLs can be considered as thin lenses in paraxial designs of achromatic systems, but in practice, the refractive lens must be treated differently. Given that the refractive indices of materials in the X-ray regime barely differ from that of vacuum, high-resolution imaging necessitates placing many refractive lenses in a row to accumulate focusing power. These compound refractive lenses (CRLs) must then be treated as thick lenses in the paraxial analysis of achromatic imaging, as has been carried out in the analysis of Poulsen *et al.* [13]. In §4, we extend and improve upon that work by noting that not only does the focal length of a CRL change with wavelength but also does the position of its principal planes. We find that this change of the location of the focal plane with a change in wavelength must be accounted for to properly describe the imaging performance of such optical systems. This is carried out using an accurate

---

[1]The power of the refractive lens can be considerably relaxed if the high dispersion of elements near their absorption edges is exploited [11]. However, the wavelength span for this is limited and we do not consider that case in this paper.

yet very tractable formalism of the paraxial optics of CRLs introduced in §3, by noting the analogy of a CRL to a thick gradient refractive index (GRIN) lens [23]. (This approach also enables the derivation of the pulse front through thick refractive lenses, given in appendix A.) The design space of achromats in §4 is parametrized in terms of product of the length and refractive gradient of the refractive lens, as well as the focal length and distance of the diffractive lens. The achievable bandwidths of thick-lens achromats are examined in §5. The high relative bandwidths found for apochromatic designs—which can exceed 10%—are verified by ray tracing in §6. Finally, some examples are presented in §7. A list of symbols used in the paper is given in table 1.

## 2. Thin lens achromats

### (a) The thin-lens doublet

An achromatic doublet lens is formed by placing two lenses in contact that have different dependences of focusing power on wavelength. In visible-light optics this is conventionally achieved by combining lenses made of different glasses. For focal lengths $f_a$ and $f_b$ of the two lenses, the achromatic condition is found when $f_a/V_a = -f_b/V_b$ where $V$ is the dispersive power given by equation (1.1). The resulting focal length is $f = f_a V_a/(V_b - V_a)$, showing that lenses of differing dispersive powers are required. All diffractive lenses have $V = -1$ as explained above and so an achromat consisting of an MLL must be combined with a refractive lens. This refractive–diffractive achromat for X-rays was proposed independently by Skinner [10] in the context of astronomy and by Wang et al. [11] for microscopy and lithography. Since $V = -2$ for X-ray refractive lenses (away from absorption edges), a diverging refractive lens of focal length $-2f_0$ at a wavelength $\lambda_0$ combined with an MLL of focal length $f_0$ at the same wavelength will give an achromat of $2f_0$ focal length with zero dispersion at $\lambda_0$. That is, a positively focusing achromatic doublet requires a negative (diverging) refractive lens placed in contact with a positive diffractive lens.

The power of the achromat doublet lens, given by the reciprocal of the focal length, is

$$\frac{1}{f_A} = \frac{1}{f_R} + \frac{1}{f_D}, \tag{2.1}$$

where $f_R = -2f_0\lambda_0^2/\lambda^2$ is the focal length of the refractive lens and $f_D = f_0\lambda_0/\lambda$ is the focal length of the diffractive lens. Expanding equation (2.1) and defining $\Delta\lambda = \lambda - \lambda_0$ gives

$$\frac{1}{f_A} = \left(1 - \left(\frac{\Delta\lambda}{\lambda_0}\right)^2\right)\frac{1}{2f_0}, \tag{2.2}$$

which obviously has zero linear dispersion at $\Delta\lambda = 0$. We define the higher-order dispersion terms $V^{(j)}$ as the coefficients of the powers of $\Delta\lambda/\lambda$ in the Taylor-series expansion of $f(\lambda)/f(0)$. Thus, we see from equation (2.2) that the achromat doublet has $V^{(2)} = 1$.

Although the phase velocity of X-rays propagating through a medium of refractive index $n = 1 - \delta$ exceeds the speed of light in the vacuum, the speed of a short pulse is given by the group velocity, $v_g = \partial\omega/\partial k = c/(n - \lambda\partial n/\partial\lambda) = c/(1 + \delta)$, where $\omega$ is the X-ray frequency and $k$ is the wavenumber, and as above, we have assumed that $\delta \propto \lambda^2$. Thus, when light propagates through different thicknesses of a material, as in a lens, the pulse front will separate from the wavefront [24]. A bi-convex lens has negative focal length in the X-ray regime with a thickness of the refractive material that is greatest on axis and reduces quadratically with distance $y$ from the axis. In this case, the pulse front lags behind the phase front on the axis and coincides with the phase front at the periphery of the lens where the thickness is zero. Relative to the pulse front on axis, therefore, the pulse leads the wavefront by a duration that increases as $y^2$, which is to say that meridional rays propagate through the lens faster than axial rays. The opposite is true for a

**Table 1.** List of symbols and their meanings.

| | |
|---|---|
| $y, z$ | Cartesian coordinates of height from the optical axis and distance along the optical axis |
| $y_0, y_0'$ | ray height and gradient at the entrance face of a refractive lens |
| $y_N$ | height of the $N$th zone in a diffractive lens |
| $\lambda, \lambda_0, \Delta\lambda$ | wavelength, wavelength for a particular design and deviation of the wavelength from $\lambda_0$ |
| $c$ | speed of light in vacuum |
| $v_g$ | group velocity of light in a medium |
| $n, n_0, \bar{n}$ | refractive index, refractive index at the optical axis and average refractive index of a lens as projected along the optical axis |
| $\delta, \delta_1, \delta_2$ | refractive index decrement ($n = 1 - \delta$), values of $\delta$ for the two materials in an MLL |
| $g, g_0$ | gradient parameter (with units of inverse length) of the refractive index profile, gradient parameter at the design wavelength |
| $R, T$ | radius of curvature and thickness of lens elements in a CRL |
| $L$ | length of the thick refractive lens |
| $P$ | lens diameter |
| NA, $NA_D$ | numerical aperture, NA of the diffractive lens |
| $\delta_r$ | image resolution ($=\lambda/(2NA)$) |
| $f$ | focal length |
| $f_0$ | focal length parameter, equal to the focal length of the diffractive lens at the design wavelength |
| $\Delta f$ | defocus (due to change in wavelength) |
| $f_A, f_R, f_D, f_{D0}$ | focal length of the achromat, the refractive lens, the diffractive lens and the diffractive lens at the design wavelength |
| $l_o, l_i$ | object and image distances |
| $F_o, F_i$ | focal planes in the object and image spaces (front focal plane and back focal plane) |
| $U_o, U_i$ | principal planes in the object and image spaces |
| $l_1, l_2, l_3$ | path lengths of rays used for calculations of time delays |
| $b, b_I, b_{II}, b_0$ | working distance between exit face of the lens and the image plane, working distances for Type I and Type II systems and at the design wavelength |
| $d$ | distance between the principal planes of the refractive and diffractive lenses |
| $D$ | distance between the faces of the two lenses |
| $s$ | distance from the exit face of a lens to its back focal plane |
| $w$ | distance from the back principal plane to the lens exit face |
| $V, V^{(2)}$ | dispersive power (see equation (1.1)) and second-order dispersive power |
| $v, v^{(2)}$ | image distance dispersion and second-order image distance dispersion |
| $\alpha_I, \alpha_{II}$ | ratio $f_R/f_0$ of the refractive lens in an achromat at the design wavelength, for a Type I and Type II system |
| $\beta_I, \beta_{II}$ | ratio $b/f_0$ of the achromat at the design wavelength, for a Type I and Type II system |
| $\gamma$ | ratio of the lens gap $D$ to the length $L$ of the refractive lens |
| $T_g, T_\phi$ | time of flight based on the group and phase velocities |
| $\Delta T, \Delta T_D, \Delta T_R, \Delta T_I, \Delta T_{II}$ | delay between the phase front and pulse front, delay for the diffractive and refractive lenses and for Type I and Type II systems |

positive (converging) lens, and in general, the delay between the phase front and the pulse front in a thin lens, due to linear dispersion, is given by [24]

$$\Delta T = \frac{-y^2}{2cf^2} \lambda \frac{\partial f}{\partial \lambda} = \frac{-y^2}{2cf} V.$$

(2.3)

Here, when $\Delta T < 0$, the propagation time is shorter for rays at $y$ than on the axis, which occurs for an X-ray refractive lens with $f < 0$ since in that case $V = -2$.

A pulse is not delayed in traversing a diffractive lens (in the limit of zero thickness), but rays brought to a focus by a positive diffractive lens accrue a wavelength of path for each period of the structure, as was noted earlier. If we consider a plane wave focused by a diffractive lens, a ray intersecting at $y$ must traverse an extra distance $y^2/(2f)$ to reach the focus at $f$, compared with the axial ray, and it will take it longer to get there. Thus, we see that equation (2.3) holds too for a diffractive lens, for which $V = -1$ [25]. Furthermore, for the achromatic doublet consisting of a thin zone plate with $f_D = f_0$ in contact with a thin negative refractive lens with $f_R = -2f_0$, the pulse front—initially ahead of the wavefront in the refractive lens—is brought back in coincidence with the wavefront by the extra path length required of the diffractive lens. The propagation delay due to the linear dispersion, $\Delta T$, is zero. This is to be expected since the achromat transports all wavelengths of the pulse to the focus without changing the relative phases of these spectral components. Also, it was shown [24] that equation (2.3) holds for any composite lens system when $f$ is replaced by the distance $l_i$ from the last lens to the image, such that $\Delta T \propto \partial l_i/\partial \lambda$. A smaller degree of stretching of the pulse may be caused by the group-velocity dispersion, proportional to $V^{(2)}$ as indicated by equation (2.2). As seen below, this too can be mostly eliminated in apochromatic designs.

## (b) Separated lenses

Skinner [12] examined the case when the diffractive and refractive lenses are separated from each other by some distance $d$ and found that this extra degree of freedom enabled the design of an apochromatic system where the quadratic dependence of the image position is brought to zero, leaving a predominantly cubic behaviour. Poulsen et al. [13] also analysed this situation in the context of using a CRL together with a diffractive lens. CRLs are required for a practical achromatic system, for the same reason they are needed for focusing and imaging—the focal length of a single refractive lens is just too long. Since the principal plane of a negative CRL is situated between the first and last lenses of the stack (as will be detailed in §3), the smallest achievable value of $d$ in this situation is greater than zero. While the focal length of the system of two thin lenses of focal lengths $f_a$ and $f_b$ can be found from the lens maker's formula as

$$\frac{1}{f_A} = \frac{1}{f_a} + \frac{1}{f_b} - \frac{d}{f_a f_b},$$

(2.4)

it is the position of the image from the lens that must not vary with wavelength. In a compound imaging system consisting of two (or more) lenses, a focal length invariant to wavelength does not necessarily imply that the image position will remain at a constant distance from the lens since the position of the back principal plane may vary with wavelength. (The back principal plane is where rays emanating back from the image would appear to intersect incident parallel rays from a source at infinity.) To choose an image plane, we thus consider the case of the source at $z = -\infty$ with the $z$-axis defining the optical axis, corresponding to a probe-based microscope. In this case, the distance of the image from the second lens $l_i$ is found via

$$\frac{1}{l_i} = \frac{1}{f_b} + \frac{1}{f_a - d},$$

(2.5)

since the intermediate focus formed by the first lens is at a distance $l_o = f_a - d$ in front of the second lens, as depicted in figure 1a. In that figure, the first lens is refractive with $f_a = f_R$ and it is diverging such that $f_a < 0$. Lengths that are negative are depicted in the figure by arrows pointing

to the left, including $l_o$ in figure 1a. Equation (2.5) holds both in this case of a diverging lens ($f_a < 0$) followed by a positive lens ($f_b > 0$) as well as the opposite case where it may be that $l_0 = f_a - d$ is positive as shown in figure 1b where $f_a = f_D$. We note that most X-ray imaging situations place either the image or object plane at close to infinity—the results presented in this paper can be applied to both cases.

As with the doublet lens (for which $d = 0$), achromatic focusing conditions can only be found for separated lenses when the refractive lens has a negative focal length and the diffractive lens is positive. There are thus two possible configurations: one where the refractive lens is followed by the diffractive lens ($f_a = f_R$, $f_b = f_D$), and the other in which these lenses are swapped ($f_a = f_D$, $f_b = f_R$). We call the first the Type I configuration and the second Type II. The ratio of the focal lengths of the refractive to the diffractive lenses at the achromatic condition for $d > 0$ is no longer $-2$, and we set $f_D = f_0\lambda_0/\lambda$ and $f_R = \alpha f_0\lambda_0^2/\lambda^2$. The achromatic condition can be found by solving for $\alpha$ in the equation

$$\left.\frac{\partial l_i}{\partial \lambda}\right|_{\lambda=\lambda_0} = 0, \tag{2.6}$$

using equation (2.5) with the appropriate choices of $f_R$ and $f_D$ for the Type I or Type II system.

For a Type I achromatic system, we obtain the solution

$$\alpha_{\mathrm{I}} = -1 + \frac{d}{f_0} \pm \sqrt{1 - \frac{2d}{f_0}}, \tag{2.7}$$

with only the choice of the minus sign giving a positive image distance $l_i$. An achromat can only be formed when $d < f_0/2$. Setting $\lambda = \lambda_0 + \Delta\lambda$ and expanding $l_i$ in a Taylor series at $\Delta\lambda = 0$ gives

$$l_i = f_0\beta\left[1 + v^{(2)}\left(\frac{\Delta\lambda}{\lambda}\right)^2 + O\left(\frac{\Delta\lambda}{\lambda}\right)^3\right], \tag{2.8}$$

where

$$\beta_{\mathrm{I}} = \frac{l_i(\Delta\lambda = 0)}{f_0} = 1 + \frac{1}{\sqrt{1 - 2d/f_0}} \tag{2.9}$$

is the ratio of the image distance to $f_0$ and

$$v_{\mathrm{I}}^{(2)} = -\frac{(\beta_{\mathrm{I}} - 3)\beta_{\mathrm{I}}}{2(\beta_{\mathrm{I}} - 1)} \tag{2.10}$$

is the quadratic dispersion coefficient of the system with respect to the position of the image plane (distinct from the dispersion $V^{(2)}$ with respect to the focal length). Equation (2.8) confirms that the system is achromatic since there is no linear dependence on $\Delta\lambda$. Together with equation (2.10), this equation also reveals the remarkable effect that the system becomes apochromatic, whereby the quadratic dispersion is nulled, at a particular separation $d$ which sets the image distance to be $3f_0$ (that is, $\beta_{\mathrm{I}} = 3$). From equation (2.9), this apochromatic condition is found when $d = 3f_0/8$ and $\alpha = f_R(0)/f_D(0) = -9/8$.

The overall focal length of the system, given by equation (2.4), is $f_A = 9f_0/4$ for this condition. This is slightly longer than the focal length of $f_A = 2f_0$ obtained for the doublet consisting of two lenses in contact. It should be noted that $\partial f_A/\partial\lambda \neq \partial l_i/\partial\lambda$ (or $V \neq v$) when the lenses are separated. A consequence of this is that while various wavelengths are brought to focus to the same image plane, the image magnification will vary with wavelength. Thus, achromatic focusing only occurs for a source located on axis. The image of an off-axis source point will be dispersed laterally (as in an aberration-free spectrometer).

A similar analysis applied to the Type II system gives a solution for the ratio of the refractive focal length to the diffractive focal length given by

$$\alpha_{\mathrm{II}} = -2\left(1 - \frac{d}{f_0}\right)^2. \tag{2.11}$$

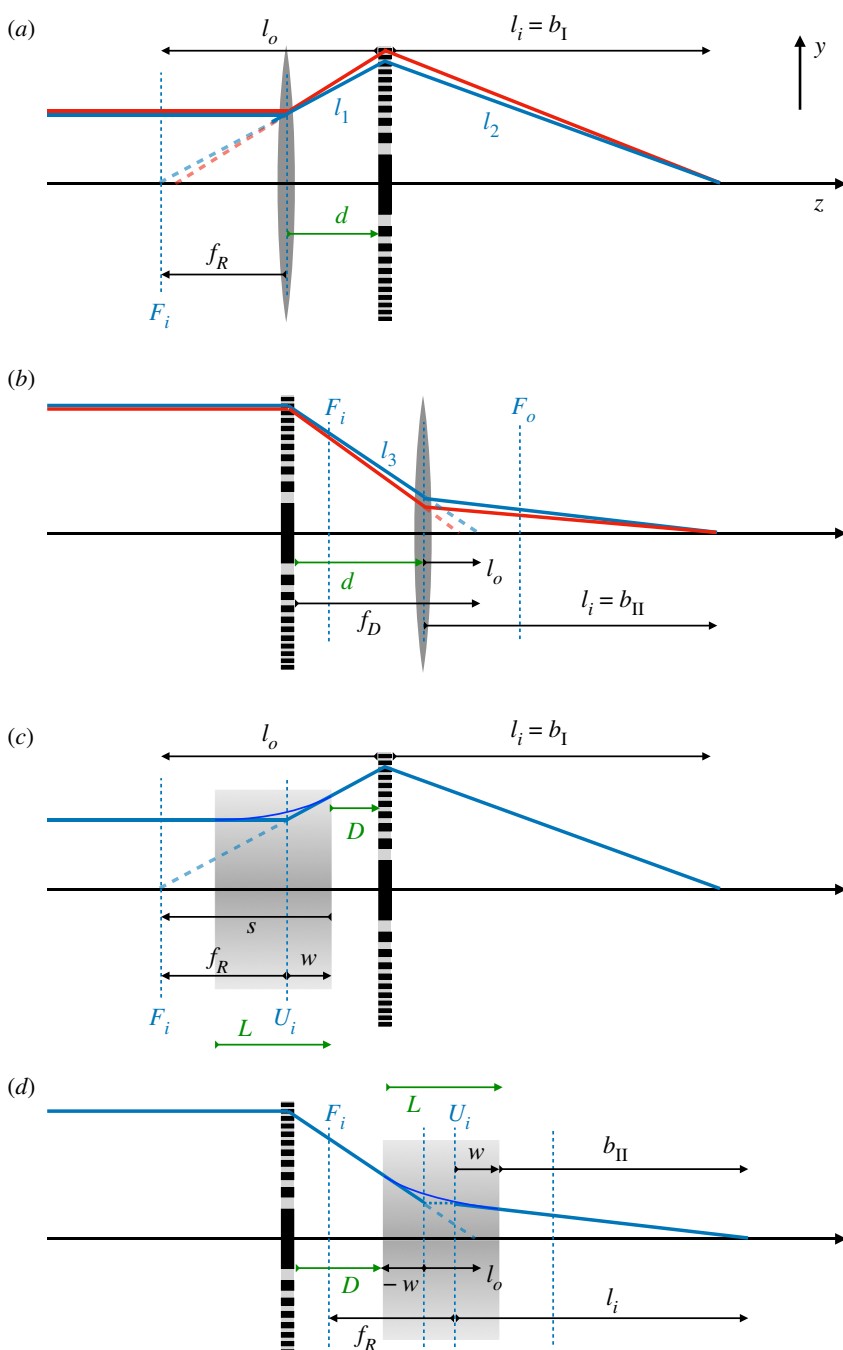

**Figure 1.** Paraxial optics analysis of achromatic systems imaging a source at $z = -\infty$. (a) Type I system consisting of thin lenses separated by $d$; (b) Type II thin-lens system; (c) Type I system consisting of a thick refractive lens (TRL) of length $L$ a distance $D$ from a thin diffractive lens; (d) Type II system consisting of a TRL and a thin diffractive lens. All distances displayed in black and blue are wavelength dependent. Arrows pointing right indicate a positive length, and left-pointing arrows indicate a negative length (e.g. $f_R < 0$). The focal planes and principal planes of the refractive lenses are shown by the blue dashed lines; the focal planes in the image and object spaces are labelled as $F_i$ and $F_o$, respectively, and the principal planes $U_i$ and $U_o$. The working distances for the Types I and II systems are $b_I$ and $b_{II}$. The refractive index of the medium of these lenses is <1. The red rays in (a) and (b) are for a wavelength that is 10% longer than for the blue rays.

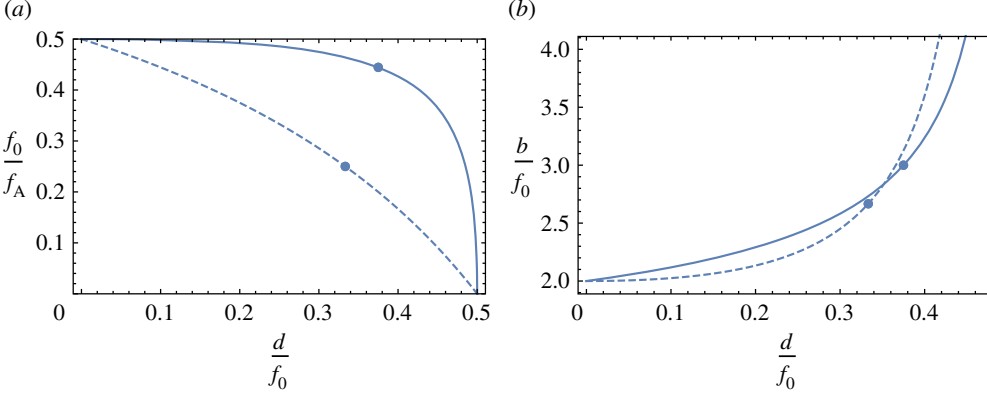

**Figure 2.** Plots of (*a*) the power and (*b*) the image distance for thin-lens achromats as a function of the separation of the lenses. All distances are normalized to the focal length of the diffractive lens, $f_0$. Type I systems are depicted with solid lines and Type II with dashed lines, and the apochromatic conditions are shown by the circles. (Online version in colour.)

It is found that the image distance as a function of $\Delta\lambda$ at the achromatic condition follows equation (2.8) with

$$\beta_{\mathbb{I}} = 2\frac{(1 - d/f_0)^2}{1 - 2d/f_0} \tag{2.12}$$

and

$$\nu_{\mathbb{I}}^{(2)} = \frac{1 - 3d/f_0}{(1 - 2d/f_0)(1 - d/f_0)}. \tag{2.13}$$

At $d = f_0/2$, it is seen from equation (2.12) that the image is formed at infinity and, as for the Type I system, a real image can only be achieved when $d < f_0/2$. The quadratic dispersion of equation (2.13) is expressed in terms of $d/f_0$ as this gives a more compact form than using $\beta$ (the opposite was true for the Type I case in equation (2.10)). It can be immediately seen that an apochromatic condition is achieved at $d = f_0/3$. In this case, $\alpha_{\mathbb{I}} = -8/9$, $\beta_{\mathbb{I}} = 8/3$ and $f_A = 4f_0$.

Plots of the relative focal lengths and image distances as a function of the normalized lens spacing $d/f_0$ are given in figure 2. The Type I system gives higher focusing power (shorter focal length) for all lens spacings, and thus is the preferable configuration for high resolution imaging. The apochromatic condition for a Type I system gives a performance almost as good as that of the doublet.

As discussed in §2a, the achromatic condition of $\partial l_i/\partial\lambda = 0$ ensures that all rays of a short pulse are brought to the focus at the same time, $\Delta T = 0$. Calculation of propagation times through the lens systems can therefore serve as an independent check of the derivations of the achromatic conditions and are given in appendix A.

## 3. Paraxial optics of thick refractive lenses

A typical value of the refractive index decrement $\delta$ of light materials (such as diamond, for example) is about $10^{-6}$ at a wavelength of 0.05 nm (a photon energy of 24.8 keV). A bi-convex lens with surfaces of radius $R$ has a (negative) focal length of $f_R = -R/(2\delta)$ which is therefore of the order of metres for lens radii of the order of micrometres. This compares with the millimetre focal lengths of high-resolution MLLs which have apertures up to about 100 µm and which we would like to pair in an achromat. Such pairing, as seen above, requires lenses with focal lengths of comparable magnitude. As is now common practise, stacking $N$ positive refractive lenses in a row along the optical axis sums their focusing powers to modify the focal length by a factor of $1/N$. The same is true for negative lenses.

It is clear that to create a high-resolution achromat, such a negative CRL will require many hundreds or even thousands of lens elements. Rays traversing this lens will not be deflected in one particular plane as was assumed for the analysis of §2 but will be gradually nudged as they pass through each element. Different wavelengths will deflect by slightly different amounts and thus follow different trajectories. In the limit of many lens elements, these trajectories will appear curved.

In paraxial optics, as a consequence of treating each lens element as a linear system, any composite lens can be assigned two principal planes, $U_o$ and $U_i$, and two focal planes, $F_o$ and $F_i$, that together describe the total linear system [26]. The curved trajectories of rays can then be ignored and instead the ray geometry can be described solely by the intersections of straight rays with these planes, indicated in figure 1c,d for the refractive lens. Collimated rays parallel to the optical axis and impinging on the front of the lens will leave the rear of the lens to converge at the image of the source on the rear focal plane $F_i$ (for a positive lens) or appear to diverge from the rear focal plane $F_i$ (for a negative lens), in both cases as if focused by a thin lens located at the rear principal plane $U_i$. That is, the incident collimated rays appear to intersect the outgoing rays at $U_i$. Likewise, rays originating from the front focal plane $F_o$ (or which would converge upon the front focal plane in the case of a negative lens) will be collimated by the composite lens, and the front principal plane $U_o$ stands at the place where each collimated ray appears to intersect with the originating ray. The two principal planes coincide with each other and the plane of the lens only when that lens is thin. As we will see below, for a negative thick refractive lens, $U_i$ is downstream of $U_o$. When the source and image are at places other than, respectively, infinity and the corresponding focal plane, the input ray that appears to intersect the front principal plane $U_o$ at a particular height ($y$, say) produces an output ray that appears to arise from the back principal plane $U_i$ at that same height $y$. Again, the entire thick lens seems to behave like a thin lens except that there is a gap between $U_o$ and $U_i$ where rays are 'teleported', or shifted along the optical axis, from one principal plane to the other (a point on $U_o$ is imaged to a point on $U_i$ with positive unity magnification).

Based on this linear-systems approach of paraxial optics, several authors have developed analyses of thick positive CRLs which can be used to describe the positions of their focal planes and principal planes. One such approach is to use the matrix transfer of vectors of ray parameters (position and direction), known as Gaussian optics. Given the transfer matrix for a single element, the analysis of $N$ equally spaced identical lenses requires evaluating its $N$th power, which can be done by diagonalizing the matrix [27,28]. In the limit of a low focusing power per lens element, each element can be treated as a matrix of differentials, leading to a set of coupled differential equations for the compound lens [29,30]. This continuous representation of a CRL mimics the behaviour of the curved trajectories of rays traversing a GRIN lens. Such a lens consists of an inhomogeneous medium where the refractive index varies continuously and quadratically with distance $y$ from the optical axis [23,31], equal to the average refractive index of the CRL as projected along the optical axis, as given by

$$n(y) = n_0\sqrt{1 + g^2y^2} \approx n_0\left(1 + \frac{g^2y^2}{2}\right), \tag{3.1}$$

where $g$ is the gradient of the refractive index (with dimensions of inverse length) and is defined here for a diverging (negative) lens where the refractive index increases with $y$. For a negative CRL composed of identical bi-convex lenses of refractive index $n_0 = 1 - \delta$, thickness $T$, surfaces of radius $R$, and without any further gap between them, the average refractive index at a height $y$ is

$$\bar{n}(y) = \left(n_0\left(T - \frac{y^2}{R}\right) + \frac{y^2}{R}\right)\frac{1}{T} = 1 - \delta + \frac{\delta y^2}{RT}. \tag{3.2}$$

Comparing this with equation (3.1) shows that $g^2 = 2\delta/(n_0RT) \approx 2\delta/(RT)$, and thus $g \propto \lambda$. We assume the refractive index profile is invariant with the coordinate $z$, equivalent to a CRL made of identical lens elements. Note that this comparison need not serve only as a simple analogy to give

a simpler analysis of CRLs, but it also shows that a thick X-ray GRIN lens (made by concurrent depositions of two materials; e.g. [32]) makes a suitable alternative to a compound lens. Here, we refer to either the CRL or GRIN lens as a thick refractive lens, abbreviated as a TRL.

The paraxial optics of GRIN lenses are well known and the continuous curved trajectories of rays can be computed by solving the ray equation [31,33]

$$\frac{\mathrm{d}}{\mathrm{d}u}\left[n(\mathbf{r})\frac{\mathrm{d}\mathbf{r}}{\mathrm{d}u}\right] = \nabla n(\mathbf{r}), \tag{3.3}$$

where $\mathbf{r}$ is the position vector of the ray and $\mathrm{d}u$ is the path element along the ray. For a diverging lens with a refractive profile of equation (3.1), ray trajectories can be written as a linear combination of solutions to equation (3.3) as

$$y(z) = A\cosh gz + B\sinh gz = C\cosh g(z - z_0), \tag{3.4}$$

where the lengths $A$ and $B$ (or $C$ and $z_0$) are determined from the position, $y_0$, and direction, $y_0'$, of the ray entering the lens at $z = 0$. Since $y'(z) = Cg\sinh g(z - z_0)$, then $z_0 = -1/g\tanh^{-1}(y_0'/gy_0)$ and $C^2 = y_0^2 - (y_0'/g)^2$. This set of solutions can be compared with those of a positive lens where the parabolic refractive profile decreases with position $y$ as $n(y) = n_0(1 - g^2y^2/2)$. In that case, the trajectories are described by sums of sines and cosines, instead of the hyperbolic sines and cosines of equation (3.4), to give rays that converge to the optical axis. For our negative lens, we consider incident rays parallel to the optical axis, whereby $y_0' = 0$ so $C = y_0$ and $z_0 = 0$, giving $y(z) = y_0\cosh gz$ and $y'(z) = y_0g\sinh gz$. Exiting the lens at $z = L$, these rays will appear to diverge from a point a distance

$$s = \frac{-1}{g\tanh gL} \tag{3.5}$$

from the rear face of the lens, as depicted in figure 1*c*. The focal length is then

$$f_R = \frac{-1}{g\sinh gL} \tag{3.6}$$

and so the distance of rear surface of the lens from the back principal plane of the lens is given by

$$w = f_R - s = \frac{\tanh(gL/2)}{g}. \tag{3.7}$$

As expected, the focal length $f_R$ is negative ($F_i$ is upstream of $U_i$), and we find the rear surface is located a positive distance $w$ from $U_i$. Since the compound lens is invariant to inversion in $z$, the front surface is located a negative distance $-w$ from $U_o$ and the front focal plane $F_o$ is located a positive distance $-f_R$ from the principal plane $U_o$. (For a positive lens $F_i$ is downstream from $U_i$ and $F_o$ upstream from $U_o$.)

For a given refractive gradient $g$, as set by the radius, thickness, and refractive index of the lens elements in the case of a CRL, the focal length reduces in magnitude as the length of the lens $L$ increases. However, the rate that the focal plane moves forward does not keep up with the increase in the length of the lens and so the principal plane $U_i$ actually moves further from the exit surface as the lens extends. The overall scale of the lens and the focal length is set by the length $1/g$, and as we will see below, this sets the scale and focal length of the achromatic system.

As seen above, $g \propto \lambda$, and hence $\partial g/\partial\lambda = g/\lambda$. The dispersion of the TRL, in terms of the position of the virtual image relative to the exit of the lens, is thus given by

$$\frac{\lambda}{s}\frac{\partial s}{\partial\lambda} = \frac{g}{s}\frac{\partial s}{\partial g} = -\left(1 + \frac{2gL}{\sinh 2gL}\right). \tag{3.8}$$

This approaches the thin-lens value of $-2$ as $L \to 0$ and tends to $-1$ as $L \to \infty$. The delay between the pulse front and the wavefront of a collimated beam focused by the negative TRL is derived in appendix A and given by equation (A 12). It is found that $\Delta T$ follows the same expression of equation (2.3) (for a thin lens) but with the focal length $f$ replaced with $s$.

# 4. Achromats utilizing thick refractive lenses

The paraxial optics formalism may seem to suggest that the analysis of the separated thin-lens achromats of §2 could apply in the case of TRLs, by setting the distance $d$ to the separation of the appropriate principal plane of the thick lens to the diffractive lens. This was essentially the assumption of Poulsen *et al.* [13], who analysed a Type II system formed by a diffractive lens and a CRL. However, that approach assumes that the dispersion of the refractive lens remains constant at $-2$, which equation (3.8) shows is not the case. We therefore modify the approach of §2 to account for a separation of the principal planes that is wavelength dependent. In the following, we avoid approximations of previous analyses by using the full analytical expressions of equations (3.6) to (3.7) for the TRL. We consider imaging systems that focus a source located at $-\infty$. We introduce the gap $D$ between the exit or entrance surface of the TRL and the diffractive lens, as shown in figure 1c,d. As previously, the diffractive lens is considered a thin lens such that its principal planes coincide with the plane of the lens. We expect in the limit $L \to 0$ that we reproduce the results of §2.

## (a) Type I systems

For the Type I system, referring to figure 1c, the (positive) distance between the rear principal plane $U_i$ of the TRL and diffractive lens is $d = D + w$. Given the negative focal length of the TRL, the negative object distance for the diffractive lens (that is, the distance to the virtual image created by the TRL) is $l_o = f_R - d = s - D$. The image working distance $b_I$, here equal to the image distance $l_i$ of the diffractive lens, is given by

$$\frac{1}{b_I} = \frac{1}{f_D} + \frac{1}{s - D}.$$ (4.1)

Using equation (3.5) and substituting $g = g_0 \lambda / \lambda_0$ and $f_D = f_{D0} \lambda_0 / \lambda$, we compute $\partial(1/b_I)/\partial\lambda$ at $\lambda = \lambda_0$ in a similar fashion to the procedure in §2. The stationary value of $1/b_I$ (and thus also of $b_I$) is then found to occur for

$$f_{D0} g_0 = \frac{(\cosh g_0 L + \gamma g_0 L \sinh g_0 L)^2}{g_0 L + \cosh g_0 L \sinh g_0 L},$$ (4.2)

where the gap between the lenses relative to the length of the TRL has been parametrized as $\gamma = D/L$.

The image position for an achromatic system obeying the condition of equation (4.2), for $\lambda = \lambda_0$, is given by

$$b_{I,0} g_0 = \frac{2(\cosh g_0 L + \gamma g_0 L \sinh g_0 L)^2}{g_0 L(2 + \gamma - \gamma \cosh 2g_0 L)},$$ (4.3)

which is positive for positive values of $\gamma$ and $L$, as long as $\cosh 2g_0 L < (2 + \gamma)/\gamma$, giving a useable achromat that creates a real focus. When the two lenses are in contact, $D = 0$, equation (4.3) simplifies to

$$b_{I,0} g_0 = \frac{\cosh^2 g_0 L}{g_0 L},$$ (4.4)

which is positive for all values of $g_0 L$.

## (b) Type II systems

In the Type II system, the refractive lens images a converging beam instead of a collimated one and thus we must consider both principal planes of this lens. The distance from the diffractive

lens to the front principal plane of the TRL is now given by $d = D + w$, as seen in figure 1d, which is equal to the expression for the Type I system. Now, however, $l_o = f_D - d = f_D - D - w$ so that

$$\frac{1}{l_i} = \frac{1}{f_R} + \frac{1}{l_o} = \frac{f_R + f_D - w - D}{f_R (f_D - w - D)}. \tag{4.5}$$

Noting further from figure 1d that $b_{\mathbb{I}} = l_i - w$, we obtain

$$b_{\mathbb{I}} = s - \frac{f_R^2}{f_D + s - D}. \tag{4.6}$$

Again, using equations (3.5) and (3.6), and the wavelength-dependent expressions for $g$ and $f_D$, the stationary value of $1/b_{\mathbb{I}}$ with respect to wavelength occurs when

$$f_{D0} g_0 = \frac{\cosh 2g_0 L + \gamma g_0 L \sinh 2g_0 L + 2\gamma g_0^2 L^2 \pm \sqrt{1 + 2\gamma \sinh 2g_0 L + 4(1 + \gamma)g_0^2 L^2}}{2g_0 L + \sinh 2g_0 L}. \tag{4.7}$$

Only the positive root gives a solution where $b_{\mathbb{I}}$ is positive. The image position for this achromatic condition at $\lambda = \lambda_0$ is then

$$b_{\mathbb{I},0} g_0 = \frac{1 - 2g_0 L \tanh g_0 L + \sqrt{1 + 4(1 + \gamma)g_0^2 L^2 + 2\gamma g_0 L \sinh 2g_0 L}}{2g_0 L + \tanh g_0 L \left(1 - \sqrt{1 + 4(1 + \gamma)g_0^2 L^2 + 2\gamma g_0 L \sinh 2g_0 L}\right)}. \tag{4.8}$$

When $D = 0$ ($\gamma = 0$), equation (4.8) simplifies to

$$b_{\mathbb{I},0} g_0 = \frac{1 - 2g_0 L \tanh g_0 L + \sqrt{1 + 4g_0^2 L^2}}{2g_0 L + \tanh g_0 L \left(1 - \sqrt{1 + 4g_0^2 L^2}\right)}, \tag{4.9}$$

which is positive for all values of $g_0 L$.

## (c) Achromatic and apochromatic conditions

Equations (4.2) and (4.7) show that the solutions for achromatic focusing depend only on the parameters $g_0 L$ and $\gamma$, and the required diffractive-optic focal length (and hence overall focal length and image distance) is proportional to the length $1/g_0$. This length thus determines the scaling of the imaging system and sets limits on the achievable resolution of the system. Of course, the zone plate or MLL must be produced with the appropriate focal length to meet that scaling, but in practise, after determining the focal lengths of the refractive and diffractive lenses, one would adjust the length $L$ of the TRL and the distance $D$ between them to achieve the achromatic condition. Graphs of $f_{D0}$ and $b_0 g_0$ are shown in figure 3 as a function of $g_0 L$ for different relative gaps $\gamma$ for Type I and Type II systems. The smallest values of $f_{D0}$ and $b_0$ are obtained when the diffractive lens is in contact with the TRL, $\gamma = 0$. This is not the doublet of §2 where the focal length of the refractive lens is $-2$ times that of the diffractive lens, since the principal planes of the lenses are still separated by $d = f_R - s$. For a Type I system, the smallest value of $b_{\mathrm{I},0}$ (which will give the highest NA for a given aperture of the diffractive lens) is then $2.233/g_0$ at a value of $L = 0.772/g_0$. For a Type II system with $\gamma = 0$, the image distance is decreased as $g_0 L$ is increased, with $b_{\mathbb{I},0} \to 1/g_0$ as $g_0 L \to \infty$. However, as will be seen below, this does not necessarily give the highest NA.

The achromatic behaviour of the lens systems can be checked by evaluating the image position $b$ as a function of the wavelength. This is obtained by expanding $b$ as evaluated in equation (4.1) or (4.6), using the wavelength-dependent expressions of the focal length of the diffractive lens $f_D$ and the working distance $s$ of the refractive lens. Setting $\lambda = \lambda_0 + \Delta\lambda$, we obtain (with the help of a symbolic mathematics program)

$$b = b_0 \left[1 + v^{(2)} \left(\frac{\Delta\lambda}{\lambda_0}\right)^2 + O\left(\frac{\Delta\lambda}{\lambda}\right)^3\right]. \tag{4.10}$$

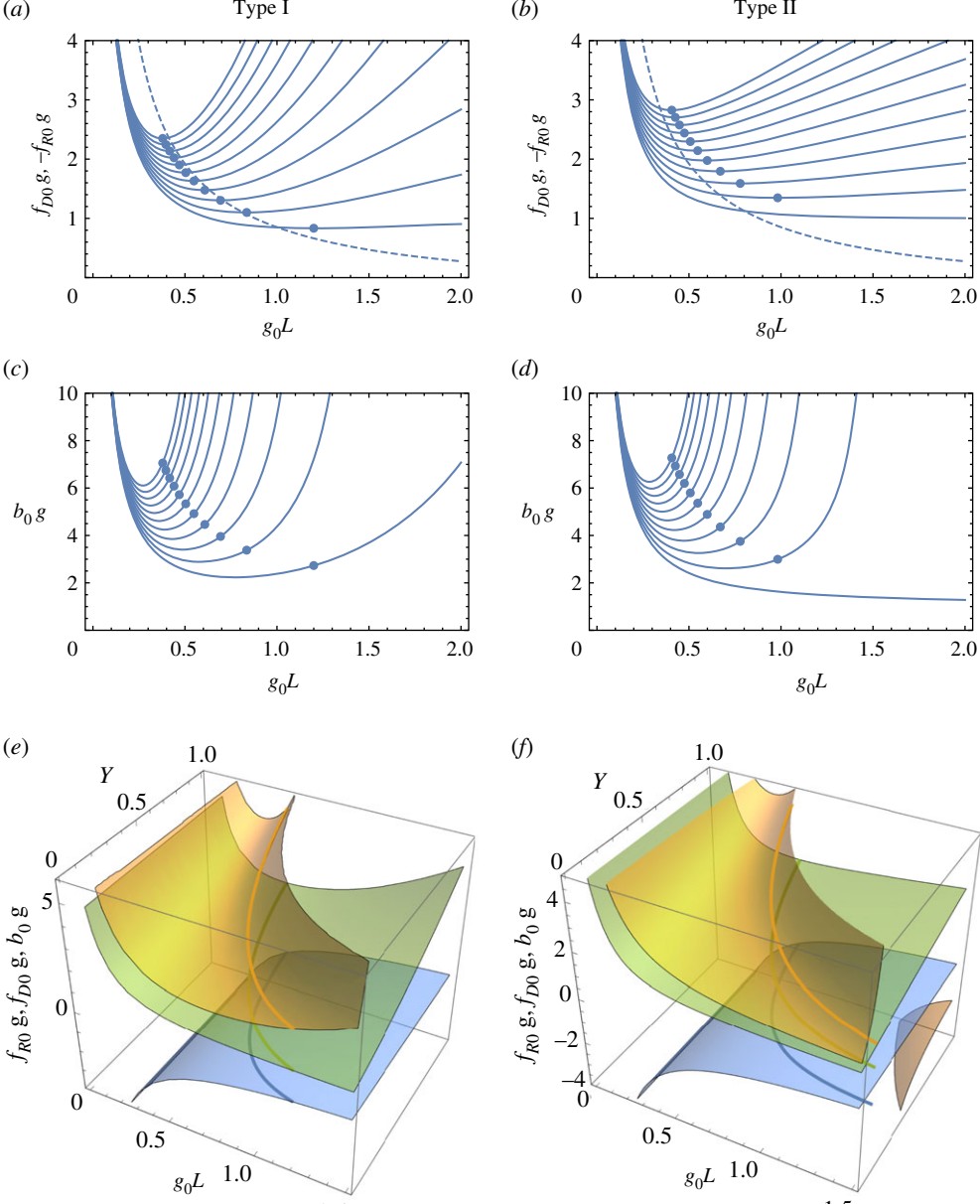

**Figure 3.** (*a,b*) Plots of the focal length of the diffractive lens (in units of $1/g_0$, solid lines) required to achieve the achromatic condition, as a function of the lens parameter $g_0L$ of the TRL, for Type I and Type II systems. The negative of the focal length of the refractive lens is shown by the dashed line. (*c,d*) The corresponding image distance $b$ (also in units of $1/g_0$). Each graph gives plots for a relative gap, $\gamma$, of 0 to 2.0 in steps of 0.2. In all four graphs, smaller $\gamma$ give smaller values of $f_{D0}$ and $b$. The apochromatic condition for each case is depicted by the blue circles. (*e,f*) The chromatic and apochromatic conditions are shown as surfaces and lines, respectively, for $f_{R0}$ (blue), $f_{D0}$ (green) and $b_0$ (orange) for Type I and Type II systems.

As expected, there is no linear dependence of $b$ on wavelength. For a Type I system, the dimensionless coefficient for the quadratic dependence on wavelength is given by

$$\nu_I^{(2)} = \frac{4g_0L(1+\gamma)\sinh g_0L + \gamma\cosh 3g_0L + ((4g_0^2L^2-1)\gamma - 4)\cosh g_0L}{2(\gamma\cosh 2g_0L - 2 - \gamma)(\cosh g_0L + g_0L\gamma\sinh g_0L)}. \tag{4.11}$$

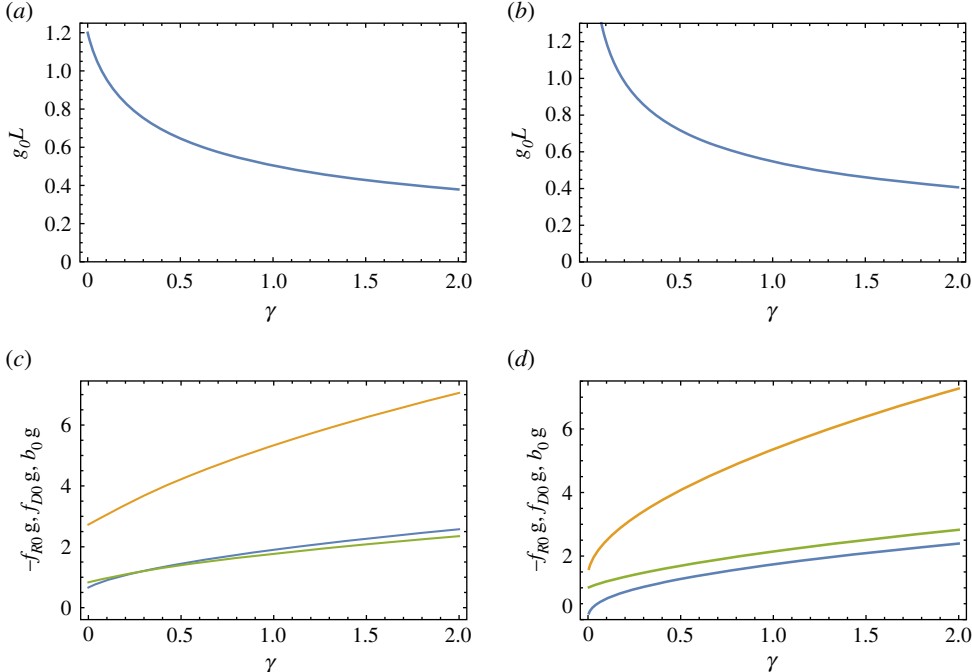

**Figure 4.** Dependence of the TRL lens parameter $g_0L$ on the relative gap of the lenses $\gamma$ for the apochromatic condition of a (a) Type I and (b) Type II lens system. The corresponding focal lengths and image distances of $-f_{R0}$ (blue), $f_{D0}$ (green) and $b_0$ (orange) in the apochromatic condition, in units of $1/g$ for a (c) Type I and (d) Type II lens system. (Online version in colour.)

For the Type II system, the expression for $v^{(2)}$ is rather long and not very illuminating and thus is not given here. Expressions of $v^{(2)}$ for both types do take on simpler forms when $\gamma = 0$, whereby

$$v_{\mathrm{I}}^{(2)} = 1 - g_0L \tanh g_0L \tag{4.12}$$

and

$$v_{\mathrm{II}}^{(2)} = -\frac{g_0L}{g_0 b_{\mathrm{II},0}}. \tag{4.13}$$

As with achromats constructed from thin lenses, the second-order dispersion $v^{(2)}$ can be made to vanish in certain situations, to give an apochromat. For lenses in contact, $\gamma = 0$, this occurs in a Type I system when $\tanh g_0L = 1/(g_0L)$ as seen from equation (4.12). This has only one solution for positive $g_0L$, which is $g_0L = 1.1997$. For other values of $\gamma$, the solution to $v^{(2)} = 0$ can be found numerically. This value of $g_0L$ for apochromatic Type I systems is plotted in figure 4a as a function of $\gamma$ between 0 and 2. The plot shows that as the gap between the lenses increases, the required setting of $g_0L$ for apochromatic imaging decreases from 1.1997. Figure 4c shows the corresponding magnitude of the focal length of the TRL, $-f_{R0}$ in blue, along with the focal length of the diffractive lens $f_{D0}$ in green and the image distance $b_0$ in orange. The smallest magnitudes of these focal lengths and image distance, achieved when $\gamma = 0$, are found to be $f_{R0} = -0.6627/g_0$, $f_{D0} = 0.8336/g_0$ and $b_{\mathrm{I},0} = 2.731/g_0$.

The Type II configuration has no apochromatic solution when $\gamma = 0$. The solution to $v^{(2)} = 0$ is plotted in figure 4b and indicates that as $\gamma \to 0$, the apochromatic condition requires $g_0L \to \infty$ and $-f_{R0} \to 0$. Any non-zero gap between the lenses—even an infinitesimal one—does give apochromatic solution, as seen in figure 4b,d.

The entire landscapes of the achromatic and apochromatic conditions are captured in figure 3e,f, where the achromatic condition gives surfaces for $f_{R0}$, $f_{D0}$ and $b_0$ as a function of the parameters $g_0L$ and $\gamma$. For any relative gap $\gamma$ there is only one value of $g_0L$ which gives the

apochromatic condition for a positive image distance $b$, shown as continuous lines in figure 3$e$,$f$ as well as by the blue circles in figure 3$a$–$d$.

As the gap $D = \gamma L$ between the two lenses is increased, figure 4$a$,$b$ shows that the required $g_0 L$ or the apochromatic condition reduces, meaning that for a given gradient $g_0$ and scaling of the optical system the length of the TRL is reduced. This may have the advantage of increasing the transmission of the TRL. The effects on the NA and resolution are examined below in §6.

## (d) Thin-lens limit

The limit of the thin lens of §2 is difficult to discern from figures 3 and 4, since we must examine the extremes of the parameter $g_0 L$ (the abscissa in figure 3) and the overall scaling by $1/g_0$ (the ordinate). The thin-lens limit certainly requires $L \to 0$ so that $w \to 0$ as per equation (3.7), ensuring that the principal planes of the refractive lens coincide. However, this limit does not give a finite focal length, since $f_R \to -1/(g_0^2 L)$ as $L$ is reduced. The combined limit of $L \to 0$, $1/g_0 \to 0$ does lead to a finite focal length—equal to zero. To recover a finite non-zero focal length we must arrange that $L$ approaches zero faster than $1/g_0$ does. Given the limiting behaviour of $f_R$, we can achieve that by setting $1/g_0 = \sqrt{-fL}$, for which $f_R \to f$ and $w \to 0$ as $L \to 0$. In addition, as $L$ is reduced the normalized gap between the lenses $\gamma = D/L$ becomes larger, and thus we see the thin-lens limit occurs in the graphs of figure 3 towards small $g_0 L$ and large $\gamma$.

## 5. Bandwidths and propagation delays of imaging systems

A change in wavelength alters the image distance $b$. The range of wavelengths that can be tolerated therefore depends on how much defocus can be tolerated in the image. For a diffraction-limited imaging system with a square aperture, the depth of focus is given by

$$\mathrm{DOF} = \frac{2\lambda}{\mathrm{NA}^2} = \frac{8\delta_r^2}{\lambda}, \tag{5.1}$$

where $\delta_r = \lambda/(2\mathrm{NA})$ is the transverse resolution. We saw that for a single lens element, the dispersion of the imaging system is given by $\Delta b/b = \nu \Delta\lambda/\lambda$ with $\nu = -1$ or $\nu = -2$ for the diffractive and refractive lens, respectively. In these cases, the required condition that $\Delta b < \mathrm{DOF}$ reduces to

$$\frac{\Delta\lambda}{\lambda} < \frac{8\lambda}{2b\,\mathrm{NA}} \frac{\lambda}{2\mathrm{NA}} \frac{1}{|\nu|} = \frac{8\delta_r}{P|\nu|}, \tag{5.2}$$

for a lens of height or diameter $P$. As was seen in the Introduction, the ratio of the resolution $\delta_r$ to the diameter is $2N$ and so, in accordance with the discussion that each of the $N$ layers adds a wavelength of optical path, the relative bandwidth is inversely proportional to the number of layers. For, say, 3 nm resolution with a lens of 200 µm diameter, equation (5.2) gives a rather stringent limitation of $\Delta\lambda/\lambda < 1.2 \times 10^{-4}$ for a diffractive lens.

The tolerable bandwidth of an imaging system relates to the shortest pulse that can be passed by that system. Ensuring that the relative bandwidth of a short pulse is no greater than $1.2 \times 10^{-4}$, by passing it through a monochromator, for example, will stretch the duration to at least about $1.7 \times 10^4 \lambda/c$. For a wavelength of 0.08 nm, this corresponds to 4.4 fs.

An achromat eliminates the linear dispersion of the optical system, and it is clear that the tolerable bandwidth increases to

$$\frac{\Delta\lambda}{\lambda} < \left( \frac{8\delta_r}{P|\nu^{(2)}|} \right)^{1/2}. \tag{5.3}$$

For the 3 nm-resolution example, the doublet lens of equation (2.2) (where $\nu^{(2)} = 1$) increases the relative bandwidth from $1.2 \times 10^{-4}$ to 1.1%. The shortest focusable pulse would be reduced from 4.4 fs to 49 as for a wavelength of 0.08 nm. Other achromats discussed above generally have even smaller quadratic dispersion, giving higher bandwidth. Of course, the apochromats, for which $\nu^{(2)} = 0$, give the largest bandwidth. For the example considered here, we would expect this to reach $(1.2 \times 10^{-4})^{1/3} = 4.9\%$ and a pulse stretching no greater than 11 as. In fact, bandwidths of up

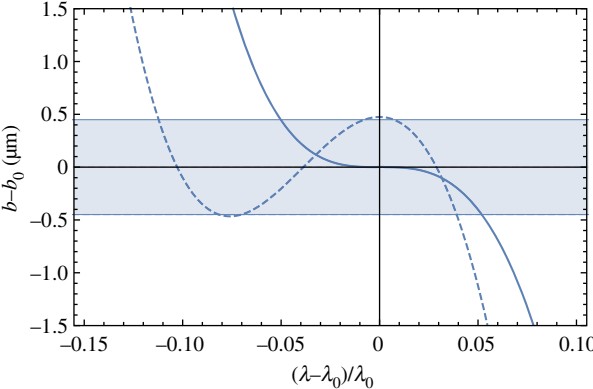

**Figure 5.** Plot of the image position as a function of relative wavelength for a Type I apochromat. The solid line shows the lens designed for the apochromat condition with $\gamma = 0$ and the dashed line has the lens separation $D$ increased to maximize bandwidth. The shaded region indicates the depth of focus for the case of a lens system with 3 nm spot size at a wavelength of 0.08 nm (15.5 keV photon energy). This lens system would not stretch a short pulse by more than 2 as. (Online version in colour.)

to 20% can be achieved by balancing some degree of linear dispersion to limit the defocus, which would allow the focusing of pulses below 2 as. A plot of the relative change in the image distance $\Delta b/b$ as a function of the relative change in wavelength $\Delta \lambda/\lambda$ is given by the solid line in figure 5 for a Type I apochromat of a TRL in contact with a diffractive lens ($\gamma = 0$). As noted above, this requires $g_0 L = 1.997$, giving $b_0 = 2.731/g_0$. It is seen that $b$ follows a cubic function, as expected, but the bandwidth can be increased by half by moving the lenses apart to a separation of $\gamma = 0.013$ to balance the dispersion with a linear term. This was optimized such that the two turning points of $b(\lambda)$ occur at the extremes of the tolerable defocus values. It is seen from figure 5 that $b(\lambda)$ can be made to equal $b_0$ at three distinct wavelengths. The shaded region in the plot of figure 5 indicates the depth of focus for a particular example of $\delta_r = 3$ nm. This could be accomplished, for example, for a TRL of 3.1 mm in length with $1/g_0 = 2.58$ mm coupled with a diffractive lens of focal length of 2.15 mm and a diameter of 120 μm.

It is worth mentioning that the relative bandwidths of most apochromat designs that can be realized for high-resolution imaging at X-ray wavelengths are not very dissimilar to the example shown here of 20%. This is because the bandwidth depends only on the cube root of the ratio of resolution to lens diameter.

# 6. Ray tracing of apochromats

The paths of meridional rays within the length of the negative TRL follow hyperbolic cosine trajectories given by equation (3.4), calculated using the wavelength-dependent refractive gradient, $g(\lambda) = g_0 \lambda/\lambda_0$. In free space, the ray trajectories are calculated as straight lines and, following the paraxial approximation to a lens, the diffractive optic modifies the ray directions according to $y' \mapsto y' - y/f_D$ using the wavelength-dependent focal length $f_D = f_0 \lambda_0/\lambda$.

Examples of calculated ray trajectories in apochromats are shown in figure 6, with the positions of the TRL and diffractive lenses depicted by the light grey and dark grey rectangles, respectively. Collimated rays are incident from the left. Rays at a long wavelength of $1.05\lambda_0$ are shown in red and rays of a short wavelength of $0.95\lambda_0$ are blue. Relative gaps of $\gamma = 0$, 0.5 and 2 are shown for the Type I systems, and 0.01, 0.5 and 2 for the Type II systems. All TRLs have the same refractive gradient $g$, but their lengths $L$ are adjusted to achieve the apochromatic condition, as are the focal lengths of the diffractive lenses. The scales of all the diagrams are consistent but the horizontal and vertical scales are not equal: the incident ray heights range from 0 to 0.1 (in units of $1/g_0$) for the Type I systems and the length of the TRL in the Type I system with $\gamma = 0$ is $1.1997/g_0$.

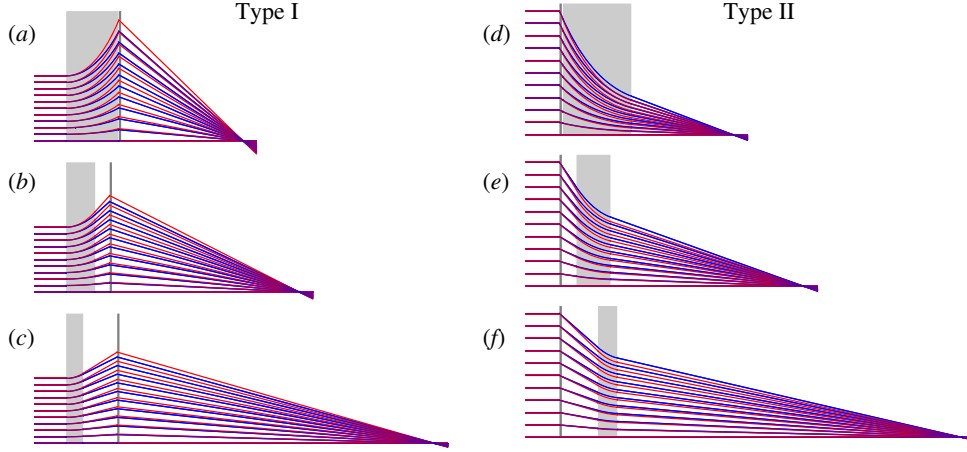

**Figure 6.** Ray trajectories of meridional rays focused by apochromatic lens systems for the same TRL gradient $g$, but different relative separations of the refractive and diffractive lenses. The red rays are traced for a wavelength of 1.05 and the blue rays for 0.95, or a bandwidth of 10%. The TRLs are depicted by light grey rectangles and diffractive lenses by darker rectangles. All images are shown on the same scale, but the horizontal and vertical scales are not equal. The incident ray heights vary from 0 to $0.1/g$, and the length of the TRL for the Type I system is $1.1997/g$.

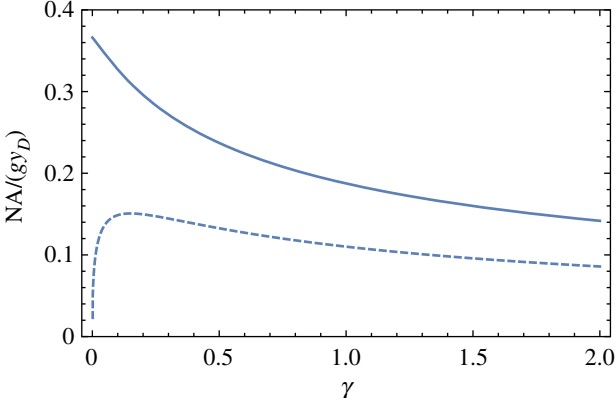

**Figure 7.** NA of Type I (solid lines) and Type II (dashed lines) apochromats, as a function of the relative gap between the refractive and diffractive lenses. The NA is plotted in units of $y_D g$.

It is apparent from figure 6 that smaller gaps between the lenses give shorter image distances and hence smaller diffraction-limited spot sizes. The NA of the Type II system is considerably less than that of the Type I, given component lenses of similar focal length and aperture size. Assuming that the limiting aperture is given by the size of the diffractive lens, the NA of a Type I system is approximately $y_D/b$ where $y_D$ is the aperture radius. For a Type II system, the extreme ray leaves the rear principal plane of the TRL at a height $y_R = y_D(f_D - D - w)/f_D$, to travel a distance $l_i = b_{\mathbb{I}} + w$ to the image plane, giving $NA = y_R/l_i$. Plots of the achievable NA of the lens system, in units of $y_D g$, are given in figure 7 for apochromatic lens systems as a function of the relative gap $\gamma$. For a given refractive gradient $g$, Type I systems give about twice the NA of Type II systems. As $\gamma \to 0$, the NA of the Type II system approaches 0. A maximum of $NA = 0.151 y_D g$ is obtained for Type II systems when $\gamma = 0.150$, and the NA falls off slowly as the gap is increased.

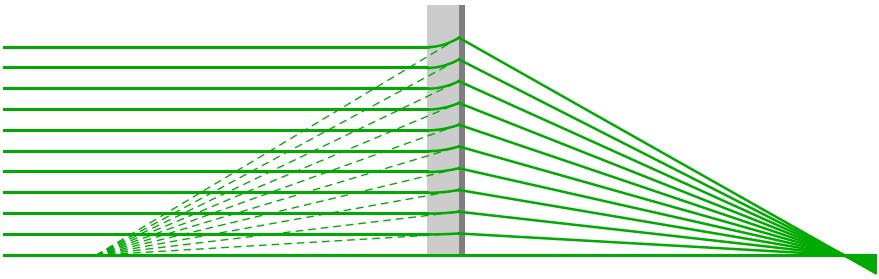

**Figure 8.** Ray trace of a Type I achromat with $gL = 0.3$ and $\gamma = 0$, for which the TRL creates a virtual image that is relayed by the diffractive lens at almost unit magnification. This requires an MLL with almost parallel layers. (Online version in colour.)

## 7. Considerations and examples using multilayer Laue lenses

For short wavelengths (high photon energies) MLLs give much higher efficiencies than zone plates fabricated by lithographic techniques. They are also able to achieve higher spatial resolution, since the layers in MLLs can be smaller than 1 nm. However to reach optimal diffracting efficiency across the entire pupil of the lens, the layers must be tilted to ensure Bragg's law is satisfied [1]. For an MLL focusing a collimated beam to a focal point a distance $f_D$ beyond the lens, the layers should all lie on the surfaces of cones (for an axisymmetric lens) or planes (for a one-dimensional lens) with a common apex located a distance $2f_D$ from the lens. This is the requirement for MLLs in Type II achromats. In a Type I system, however, the MLL forms an image of the virtual image created by the TRL, a distance $-l_o = d - f_R$ from the MLL. In this case, the layers of the MLL must lie on cones or planes that converge at a point upstream of the MLL a distance $R_C$ from the lens, found from $1/(2R_C) = 1/l_o + 1/l_i$. An interesting case is therefore when $-l_o = l_i = b_{I,0} = 2f_{D0}$ since then all the layers in the MLL must be parallel to each other and the optical axis. A comparison of equations (4.2) and (4.3) yields

$$\frac{b_{I,0}}{f_{D0}} = \frac{2g_0 L + \sinh 2g_0 L}{g_0 L(2 + \gamma - \gamma \cosh 2g_0 L)}, \tag{7.1}$$

which approaches 2 as $g_0 L \to 0$ (the thin-lens solution) for any value of $\gamma$, but approaches this value faster as $\gamma$ is reduced. Such a system can make a non-wedged MLL useful for imaging at high resolution (and achromatic) by the addition of a TRL, at the cost of doubling the image working distance and the achievable spot size. As an example, consider a negative TRL of a length $L = 0.3/g_0$ which has a focal length of $-1.015/g_0$. The achromat condition requires $f_{D0} = 1.767/g_0$. With the lenses in contact ($\gamma = 0$) equation (4.3) gives $b_{I,0} = 3.642/g_0$, which is 2.06 times the focal length of the MLL. A ray trace of this lens is given in figure 8.

The Bragg condition depends on wavelength, and thus for a given layer or zone structure of an MLL the tilt of the layers is only optimized for one particular wavelength. There is therefore a question as to whether all rays within the bandwidth of an apochromat will be efficiently focused by the MLL. All wavelengths within the bandwidth of the achromat will converge on the image point at $b$, but they will appear to have originated from secondary source points at different distances from the lens. The largest range of wavelengths that can be accepted by the MLL depends on the rocking-curve width of the Bragg reflection for the thinnest layers, which in turn depends on the wavelength. For a Type I system, all rays impinging on the image point from a certain direction will, obviously, diffract from the same point in the diffractive lens, for all wavelengths in the bandwidth. By reciprocity, the range of wavelengths that are diffracted is related to the angular dispersion of a collimated beam. From Bragg's law, this is $\Delta\lambda/\lambda = \Delta\theta/\tan\theta \approx \Delta\theta/\theta$, where $2\theta$ is the angle that rays are deflected by diffraction. Following dynamical diffraction theory applied to multilayer structures, it can be shown that $\Delta\theta/\theta \approx |\delta_1 - \delta_2|/\theta^2$, where $\delta_1$ and $\delta_2$ are the decrements of the refractive indices of the two

**Table 2.** Examples of Type I apochromat designs with $\gamma = 0$, with refractive indices calculated for diamond. The resolution is computed for a diffractive lens of 100 $\mu$m radius.

| $E$ | $\delta$ | $1/g$ | $f_D$ | $f_R$ | $f_A$ | $w$ | $L$ | $b$ | $\delta_r$ | DOF |
|---|---|---|---|---|---|---|---|---|---|---|
| (keV) | ($10^{-6}$) | (mm) | (mm) | (mm) | (mm) | (mm) | (mm) | (mm) | (nm) | ($\mu$m) |
| 8.0 | 11.41 | 1.32 | 1.10 | −0.88 | 2.00 | −0.71 | 1.59 | 3.62 | 2.80 | 0.41 |
| 15.5 | 3.00 | 2.58 | 2.15 | −1.71 | 3.90 | −1.39 | 3.10 | 7.05 | 2.82 | 0.80 |
| 40.0 | 0.45 | 6.67 | 5.56 | −4.42 | 10.06 | −3.58 | 8.00 | 18.21 | 2.82 | 2.06 |
| 100 | 0.073 | 16.55 | 13.80 | −10.97 | 24.98 | −8.89 | 19.86 | 45.22 | 2.80 | 5.07 |
| 500 | 0.0029 | 83.05 | 69.23 | −55.03 | 125.33 | −44.59 | 99.63 | 226.88 | 2.81 | 25.53 |
| 15.5 | 3.00 | 8.16 | 6.81 | −5.41 | 12.32 | −4.38 | 9.80 | 22.31 | 8.92 | 7.96 |

materials that make up the multilayer structure [2,34]. The narrowest rocking curve therefore occurs at the largest scattering angle, where $2\theta = NA_D$, the NA of the diffractive lens, giving

$$\frac{\Delta\lambda}{\lambda} \approx |\delta_1 - \delta_2| \frac{4}{NA_D^2}. \tag{7.2}$$

As an example, for an MLL made from SiC and WC layers, $|\delta_1 - \delta_2| = 6.7 \times 10^{-6}$ at a wavelength of 0.08 nm, so that $\Delta\lambda/\lambda = 1.2\%$ when $NA_D = 0.046$, corresponding to an MLL of radius 100 $\mu$m and focal length 2.15 mm. When used in a Type I achromat, a resolution of 2.8 nm can be achieved, as seen in table 2. From equation (7.2), it can be seen that since $\delta_1$ and $\delta_2$ are inversely proportional to wavelength, and the NA is proportional to wavelength for a given resolution, the bandwidth for a particular multilayer material pair is proportional to the square of the resolution, $\delta_r$. Type I apochromats with 10% bandwidth are therefore limited to a resolution of about 8 nm when constructed using SiC/WC MLLs. (Other material pairs used for MLLs in this wavelength regime likely give similar results.)

MLLs are often fabricated by depositing layers onto a flat substrate, in which case they focus only in one direction like a cylindrical lens. Two orthogonally oriented lenses can then produce a two-dimensional focus. The aberrations of such systems has recently been studied [2]. Achromats and apochromats can be made in the same way either by combining a single axisymmetric TRL with two crossed MLLs or by using a separate one-dimensional achromatic lens system for each focusing dimension. In the former case, the two MLLs should ideally be in contact. If not, then the value of $\gamma$ would be different for each, requiring different focal lengths of the two MLLs. These will not, however, focus the beam to the same image plane. It may be possible to make an anamorphic TRL to compensate this, but this may be somewhat complicated. Two separate lens systems can be positioned to image to the same plane. For example, the entire length of the Type I system with $\gamma = 0$ is less than the working distance of a Type I or II system with $\gamma = 0.5$. More combinations are possible when the gradients of the two TRLs are not equal.

Some design examples are given in table 2 of thick-lens apochromatic systems with $\gamma = 0$, the case depicted in figure 6a. It was seen above that the lens systems scale with the refractive gradient, $g$, which ideally should be as large as possible. At a wavelength of 0.08 nm (15.5 keV photon energy), the refractive index decrement of diamond is $\delta = 3 \times 10^{-6}$. Constructing a diamond CRL with bi-convex lenses of radius $R = 20 \mu$m and height $h = 1 \mu$m gives $1/g_0 = 2.58$ mm. The smallest possible lens distances for a Type I apochromatic system (at $\gamma = 0$) is then $b_{I,0} = 2.731/g_0 = 7.05$ mm, $f_{D0} = 0.834/g_0 = 2.15$ mm and $f_{R0} = -0.663/g_0 = -1.71$ mm. The length of the CRL is $L = 1.1997/g_0 = 3.10$ mm. If the radius of the CRL is 20 $\mu$m (matching the radius of curvature) then the NA of the focused beam is $0.020/7.05 = 0.0028$, providing a spot size of $\delta_r = 14.1$ nm. However, if the aperture of the CRL is increased to 100 $\mu$m, then a resolution of $\delta_r = 2.8$ nm would be achieved with $NA = 0.014$. This could be realized in a CRL since the parabolic profile $y^2/(2R)$ of the lens elements can be continued to $y > R$ (e.g. [35]).

Considering the design of achromatic lens systems at harder X-ray energies, we note that the gradient $g$ for a particular material and construction scales linearly with wavelength, and thus the focal lengths and image distance for lenses in the apochromatic condition scale inversely with wavelength. Thus, the achievable diffraction-limited spot size, dependent on the ratio of the wavelength to the NA, remains constant. However, as the wavelength (and NA) is reduced the depth of focus increases inversely with wavelength for a given spot size. This means that $\Delta b/b$ remains about the same for a given imaging resolution, and thus so too does the relative bandwidth $\Delta\lambda/\lambda$.

The concept is suitable for very high photon energies. Table 2 shows examples for 100 keV and 500 keV. The latter requires a TRL length of 99.6 mm if made from the same diamond material and with the same parameters as considered above. However, at these photon energies the absorption of materials is vastly reduced, making other materials suitable for the task. For example, at 500 keV, Mo gives an increase in $g$ by a factor of 1.6 times compared with diamond. This reduces the focal lengths, which in turn decreases the achievable resolution to 2.8 nm/1.6 = 1.8 nm. The design for Mo requires a TRL of 62 mm length. The attenuation length of Mo at 500 keV is 104 mm, so the lens has a transmission of 55%. A similar resolution and higher transmission can be achieved with Cu.

# 8. Conclusion

While it is generally well appreciated that diffractive lenses such as zone plates and MLLs exhibit a strong dependence of focal length on wavelength and, relatedly, an increase in the duration of short pulses due to the differences in path lengths of rays propagating from the lens to the focus, it is perhaps not as well known that such effects are even greater in refractive lenses. This result can be surprising, given that focusing in a refractive lens can be explained as a consequence of Fermat's principle of least time. Instead, due to the variation of refractive index with wavelength (which is strong in the X-ray regime), a short pulse will take longer to reach the focus of a positive refractive lens when traversing the outer edge of the lens than as along the axis. However, the different behaviours of diffractive and refractive lenses allow systems to be constructed where the dispersion of one lens is offset by the dispersion of the other. Since the dispersion of a refractive lens is twice that of a refractive lens, this requires a negative power that is half that of the diffractive lens (that is, a negative lens with twice the focal length of the diffractive lens), to yield a system that has a residual positive focusing effect. In this case the meridional rays travel faster in the refractive lens than the axial rays (see appendix A) to compensate the time lost along the longer path length of the diffractive lens.

We exhaustively explored the design space of achromatic systems consisting of a refractive and a diffractive lens, using a paraxial analysis. Two lenses give enough degrees of freedom to find both achromatic and apochromatic designs. Apochromaticity is defined as when both the linear and quadratic dependencies of the image position on wavelength are removed, to leave a cubic dependence that can give three distinct wavelengths that are focused to exactly the same plane. Additional degrees of freedom in the design space could be introduced by adding a third lens (such as a positive diffractive lens surrounded by refractive lenses of lower power) but such schemes lead to greater complexity and lower efficiency. It is also possible to increase the design space by changing the dispersion of the refractive lens by operating near an absorption edge of the refractive material [11]. We did not explore that case here, but this is attractive when bandwidths are limited to less than about 0.2%. In our study we showed that achromatic imaging could be achieved at high resolution (spot sizes considerably below 10 nm) over a relative bandwidth of about 1%. Apochromatic imaging extends this to up to 20%, but only if the rocking-curve width of the diffractive lens allows. In such designs, pulses as short as 2 as could be focused to a 3 nm spot size without significant distortion of the pulse in time, for a mean X-ray wavelength of 0.08 nm (15.5 keV photon energy).

The systems analysed here give an image position that is stationary with wavelength, but, except for the thin-lens doublet, their focal lengths do vary considerably with wavelength. (The

difference between focal length and image distance was discussed in §2b.) This implies that the magnification of the image is wavelength dependent, which leads to a transverse dispersion of the image of an extended source or of the image of a point source that is displaced from the optical axis. For imaging at 3 nm imaging, for example, the angular misalignment of the source should be less than about 10 μrad.

There are two topologies of the positive-focus two-lens achromat designs. The Type I configuration consists of a negative refractive lens followed by a positive diffractive lens, and a Type II has the order of lenses reversed. Type I systems have the advantage of achieving higher NA for a given lens size. The configuration giving the shortest focal length and highest NA is a Type I system in which the refractive lens is in contact with the diffractive lens. However, any realizable system for high-resolution imaging must necessarily be made with a refractive lens that is thick in order to achieve the required focal length for the design that pairs with the short focal length of the diffractive lens. In the X-ray regime, this then requires a negative CRL composed of many biconvex thin lenses. This system cannot be treated as a thin lens in the analysis of an achromatic system. Not only does the focal length of the CRL vary with wavelength, but also does the position of the principal plane of the lens. Our analysis accounts for a change in distance between the principal planes of the refractive and diffractive lenses as a function of wavelength by applying the well-established paraxial optics formalism of GRIN lenses. We are not aware if it has previously been pointed out, but cylindrical GRIN lenses composed of a material whose refractive index varies quadratically with radius are equivalent to CRLs in the limit of a large number of lenses. The achromatic lens systems we present can be constructed using either CRLs or X-ray GRIN lenses, which we refer to as thick refractive lenses.

Data accessibility. This article has no additional data.

Authors' contributions. Both authors conceived the study and wrote the paper. H.N.C. carried out the analysis and visualization.

Competing interests. There are no competing interests.

Funding. The Gottfried Wilhelm Leibniz Program of the Deutsche Forschungsgemeinschaft; the European Research Council under the European Union's Seventh Framework Program ERC Synergy grant no. 609920 'AXSIS'; and the Cluster of Excellence 'CUI: Advanced Imaging of Matter' of the Deutsche Forschungsgemeinschaft – EXC 2056 (390715994).

Acknowledgements. We thank DESY (Hamburg, Germany), a member of the Helmholtz Association HGF, for support.

# Appendix A. Propagation times of composite lens systems

## (a) Thin-lens achromats

The derivations of the achromatic conditions can be confirmed by calculating the propagation times of rays through systems consisting of lenses separated by $d$. For a Type I system the delay of the pulse front relative to the wavefront is found from the sum of the negative delay of the pulse front in the refractive lens and the positive delay due to the extra path length of the diffractive lens. The delays for each of the lenses are given by equation (2.3) with the appropriate dispersion, such that

$$\Delta T = \Delta T_R + \Delta T_D = \frac{y_R^2}{c f_R} + \frac{y_D^2}{2 c f_D}, \tag{A 1}$$

where $y_R$ is the height of the ray at the refractive lens and $y_D$ is the height of the same ray at the diffractive lens. Even though the diffractive lens does not focus a collimated beam, as was the assumption in the derivation of equation (2.3), this equation still holds. This can be seen from the fact that the additional path length from the wavefront incident on the diffractive lens and from that lens to the wavefront converging onto the focus is given by

$$\Delta l_1 + \Delta l_2 = \frac{y_D^2}{-2l_0} + \frac{y_D^2}{2l_i} = \frac{y_D^2}{2 f_D}. \tag{A 2}$$

Noting that $y_D = (1 - d/f_R)y_R$, equation (A 1) then becomes

$$\Delta T_{\mathrm{I}} = \frac{y_R^2}{c}\left(\frac{1}{\alpha f_0} + \frac{1}{2f_0}\left(1 - \frac{d}{\alpha f_0}\right)^2\right)$$

$$= \frac{y_R^2}{2c\alpha^2 f_0}\left(\frac{d^2}{f_0^2} - 2\alpha\frac{d}{f_0} + \alpha(\alpha + 2)\right),$$

(A 3)

for which $\Delta T_{\mathrm{I}} = 0$ has the same solution as given by equation (2.7).

For the Type II system, $y_R = y_D(1 - d/f_D)$, so that equation (A 1) becomes

$$\Delta T_{\mathrm{II}} = \frac{y_D^2}{c}\left(\frac{1}{\alpha f_0}\left(1 - \frac{d}{f_0}\right)^2 + \frac{1}{2f_0}\right)$$

$$= \frac{y_D^2}{2c\alpha f_0}\left(2\frac{d^2}{f_0^2} - 4\frac{d}{f_0} + \alpha + 2\right).$$

(A 4)

This expression evaluates to zero for the solution given by equation (2.11).

## (b) Thick refractive lenses

The delay of rays traversing a negative TRL can be determined by calculating the time of flight along trajectories $C$ of rays by integrating over arc length elements $du$ as

$$T_g = \int_C \frac{1}{v_g(\mathbf{r})}\,du = \int_0^L \frac{1}{v_g(\mathbf{r}(z))}\left|\mathbf{r}'(z)\right|\,dz,$$

(A 5)

where the group velocity $v_g$ in the inhomogeneous material obeys

$$\frac{1}{v_g} = \frac{1}{c}\left(n - \lambda\frac{\partial n}{\partial\lambda}\right) = \frac{n_0}{c}\left(1 - \frac{g^2 y^2}{2}\right),$$

(A 6)

for the refractive index profile given in equation (3.1). For collimated rays incident on the TRL parallel to the optic axis, the trajectories are

$$\mathbf{r}(z) = (y_0\cosh gz, z),$$

(A 7)

for which, in the paraxial approximation,

$$\left|\mathbf{r}'(z)\right| \approx 1 + \frac{g^2 y_0^2}{2}\sinh^2 gz.$$

(A 8)

Therefore,

$$T_g = \frac{n_0}{c}\int_0^L\left(1 - \frac{g^2 y_0^2}{2}\cosh^2 gz\right)\left(1 + \frac{g^2 y_0^2}{2}\sinh^2 gz\right)\,dz$$

$$\approx \frac{n_0}{c}\int_0^L\left(1 - \frac{g^2 y_0^2}{2}\right)\,dz$$

$$= \frac{n_0}{c}\left(L - \frac{g^2 y_0^2}{2}L\right).$$

(A 9)

Meridional rays travel faster through the lens than the axial ray. The difference of the propagation of a pulse to the phase front can be determined by making a similar line integral of the optical

path $n\mathbf{r}(z)$ through the lens, as

$$T_\phi = \frac{n_0}{c} \int_0^L \left(1 + \frac{g^2 y_0^2}{2} \cosh^2 gz\right)\left(1 + \frac{g^2 y_0^2}{2} \sinh^2 gz\right) dz$$

$$\approx \frac{n_0}{c} \int_0^L \left(1 + \frac{g^2 y_0^2}{2} \cosh 2gz\right) dz$$

$$= \frac{n_0}{c} \left(L + \frac{g y_0^2}{4} \sinh 2gL\right), \tag{A 10}$$

so that

$$\Delta T_R = T_g - T_\phi = -\frac{g y_0^2}{4c}\left(2gL + \sinh 2gL\right), \tag{A 11}$$

where we have made a further approximation that $n_0 \approx 1$. Equation (A 11) can be expressed in terms of the transverse coordinate at the exit of the lens, $y_i = y_0 \cosh gL$, so that

$$\Delta T_R = -\frac{g y_i^2}{2c} \tanh gL \left(1 + \frac{2gL}{\sinh 2gL}\right)$$

$$= -\frac{y_i^2}{2cs^2} \lambda \frac{\partial s}{\partial \lambda}, \tag{A 12}$$

where the last equality follows from equations (3.5) and (3.8). This expression for the pulse front delay is in agreement with the result of Bor [24] as given by equation (2.3) when $f$ is replaced with the distance $s$ of the back focal plane to the lens exit. $\Delta T_R \to 0$ as $L \to 0$ and $\Delta T_R \to -y_i^2 g/(2c)$ as $L \to \infty$.

## (c) Thick-lens achromats

Using the result of equation (A 12) the pulse front delay in a Type I system can be written as

$$\Delta T_I = -\frac{g y_R^2}{2c} \tanh gL \left(1 + \frac{2gL}{\sinh 2gL}\right) + \frac{y_D^2}{2cf_D}, \tag{A 13}$$

where $y_R = y_i$ is the height of the ray at the exit of the TRL and, from figure 1c, $y_D = (1 - D/s)y_R$ is the height of the ray on the diffractive lens. That is,

$$\Delta T_I = \frac{y_R^2}{2c}\left(-g \tanh gL \left(1 + \frac{2gL}{\sinh 2gL}\right) + \frac{1}{f_D}\left(1 + Dg \tanh gL\right)^2\right). \tag{A 14}$$

The solution to $\Delta T_I = 0$ is the same as the expression given by equation (4.2).

The calculation of the delay through the TRL in a Type II achromatic system is not so amenable to analysis. Numerical simulations confirm that the delay in this system is given by equation (2.3) with the focal length replaced by the image distance from the lens exit, $b$.

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
