## [Peer Review File · Proceedings. Mathematical, Physical, and Engineering Sciences]

Review History

RSPA-2021-0334.R0 (Original submission)

Review form: Referee 1

Is the manuscript an original and important contribution to its field?

Good

Is the paper of sufficient general interest?

Marginal

Is the overall quality of the paper suitable?

Good

Can the paper be shortened without overall detriment to the main message?

Yes

Do you think some of the material would be more appropriate as an electronic appendix?

No

Do you have any ethical concerns with this paper?

No

Recommendation?

Accept with minor revision (please list in comments)

Comments to the Author(s)

The authors present a detailed design and analysis of achromatic doublet lenses at hard X-ray wavelengths, in particular also considering the stretching effect of such lenses on ultrashort X-ray pulses. Here, the authors place emphasis on the thorough treatment of refraction through extended compound refractive lenses, which previously had not been done. Based on this analysis, the authors also present a few helpful examples.

The paper is well written, and the results are presented in a clear fashion. The application of this research lies in broadband and ultrafast hard X-ray science, a field which is rapidly developing. The present work has the potential to add to this development and may therefore become more relevant in the coming years. However, the paper is clearly written with this community in mind, and the overall situation of the field is not well introduced. This makes the context of the paper harder to understand for readers without in-depth knowledge of the state-of-the-art of attosecond free electron lasers. Furthermore, I have listed a few remarks and questions below. Assuming that the authors address these, I recommend publication in Proceedings of the Royal Society A.

- The achromatic optics of obvious interest for the focusing of attosecond pulses in the hard X-ray range. In my current understanding, this is an active field of research with frequent advances in the achievable pulse duration. It would be interesting if the authors can comment on the availability of such pulses and thereby provide a context in which these optics would be of value. Would these optics still provide a clear benefit for "longer" X-ray pulses, where the requirement on pulse stretching is less stringent?
- An obvious next lens design would be one where the diffractive lens is placed "inside" of the compound refractive lens. I realize that in this case, the mathematical description becomes more complex, but is it necessarily the case that this also leads to lower efficiency, as is claimed in the conclusion? Do the authors perhaps also foresee some benefit to this arrangement?
- In the current study, analysis was limited to the focusing of collimated X-ray beams. Could similar optics also be used to image and possibly (de)magnify a prior focus?
- The authors discuss that MLLs are often fabricated as cylindrical lenses. The consequences of such a fabrication approach are discussed, but I find this discussion a bit confusing. In the situation where two MLL's are crossed and stuck together, does this have any implications for the focusing and aberrations? The use of a pair of cylindric achromats is proposed as solution for when the MLL's cannot be brought in contact. Does this solution then have any implications for the resulting focus? I would expect that this may lead to ellipticity or even astigmatism in the focus.
- In the conclusion, the authors write about the described optical systems that their "focal length do vary considerably with wavelength", however also that the image position is constant. Considering the composite lens as a single element, this appears to be self-contradictory. Is this because not only the focus length but also the principle plane shifts with wavelength?
- One of the proposed applications of this lens is ptychography. There are of course many different implementations of this technique with varying requirements, but generally, it is thought that a structured probe beam is preferred above a diffraction-limited beam (e.g. Odstrčil, Opt. Express 27(10), 2019). The benefit of structured beams is even greater if one aims to reconstruct individual spectral components of a broadband spectrum. Is the pulse stretching of other, 'structured' focusing optics so significant that these achromatic doublets provide a clear advantage that really offsets the lack of structured focusing?

Some details:

- Attosecond pulses are routinely generated in the soft-X-ray and extreme ultraviolet regimes, however the presented solution will likely not work because of increased absorption. The authors might consider adding the distinction "hard X-ray" to the title to clarify this.
- The x-axis and it's labeling in Fig. 5 are not self-explanatory; do the authors mean something like " $(\lambda - \lambda_0) / \lambda_0$ "?
- In table 1, several illuminating examples are given. For clarity, it would be beneficial to mention there also which materials were chosen for each example.

- The used symbols are generally chosen intuitively and are generally easy to remember. However, given the large number of different symbols, the authors might consider adding a list of symbols and their explanation.
- Page 3, line 53, 54 (“... and the consequent ...”): this sentence is hard to understand.

Review form: Referee 2

Is the manuscript an original and important contribution to its field?

Excellent

Is the paper of sufficient general interest?

Good

Is the overall quality of the paper suitable?

Excellent

Can the paper be shortened without overall detriment to the main message?

Yes

Do you think some of the material would be more appropriate as an electronic appendix?

No

Do you have any ethical concerns with this paper?

No

Recommendation?

Accept with minor revision (please list in comments)

Comments to the Author(s)

The authors present a theoretical analysis on the combination of diffractive and refractive lenses to achieve achromatic focusing in the X-ray regime. They thoroughly investigate several focusing solutions exploring different possible designs. They also show that with the appropriate separation between the refractive and diffractive lens, an apochromatic system can be built. The authors also demonstrate that such systems could be used to focus short X-ray pulse without distorting them in time by more than a few attoseconds, with very clear applications in temporal resolution investigations in X-ray free electron lasers. The manuscript is very well written and it presents the topic and findings in a very clear manner.

I recommend the publication of the manuscript after addressing the minor particular points of the following list:

- 1) The dispersive power, V , could already be defined in the introduction section in advance of section 2.
- 2) In page 4, line 13, is the chromatic condition missing a negative sign?
- 3) Figure 1 could be separated in two, splitting the diagrams for thin and thick lenses that actually belong to different sections of the manuscript.
- 4) Some of the labels in figure 1, such as F_i or $b(I)$ should be better clarified in the text or figure caption. The principal planes could also be displayed for the thin lenses for comparison with the thick lenses.
- 5) Typo in caption of figure 3, “Landscapes of the achromatic...”
- 6) Typo and space is missing in Page 19, line 29 “theCTRL ...”
- 7) In Figure 8, rays of two different colors could be used to display two different wavelengths.

Decision letter (RSPA-2021-0334.R0)

02-Jun-2021

Dear Dr Chapman,

On behalf of the Editor, I am pleased to inform you that your Manuscript RSPA-2021-0334 entitled "High-resolution achromatic X-ray optical systems for broad-band imaging and focusing attosecond pulses" has been accepted for publication subject to minor revisions in Proceedings A. Please find the referees' comments below.

The reviewer(s) have recommended publication, but also suggest some minor revisions to your manuscript. Therefore, I invite you to respond to the reviewer(s)' comments and revise your manuscript. Please note that we have a strict upper limit of 28 pages for each paper. Please endeavour to incorporate any revisions while keeping the paper within journal limits. Please note that page charges are made on all papers longer than 20 pages. If you cannot pay these charges you must reduce your paper to 20 pages before submitting your revision. Your paper has been ESTIMATED to be 24 pages. We cannot proceed with typesetting your paper without your agreement to meet page charges in full should the paper exceed 20 pages when typeset. If you have any questions, please do get in touch.

It is a condition of publication that you submit the revised version of your manuscript within 7 days. If you do not think you will be able to meet this date please let me know in advance of the due date.

To revise your manuscript, log into <https://mc.manuscriptcentral.com/prsa> and enter your Author Centre, where you will find your manuscript title listed under "Manuscripts with Decisions." Under "Actions," click on "Create a Revision." Your manuscript number has been appended to denote a revision.

You will be unable to make your revisions on the originally submitted version of the manuscript. Instead, revise your manuscript and upload a new version through your Author Centre.

When submitting your revised manuscript, you will be able to respond to the comments made by the referee(s) and upload a file "Response to Referees" in Step 1: "View and Respond to Decision Letter". You can use this to document any changes you make to the original manuscript. In order to expedite the processing of the revised manuscript, please be as specific as possible in your response to the referee(s).

IMPORTANT: Your original files are available to you when you upload your revised manuscript. Please delete any redundant files before completing the submission process.

When uploading your revised files, please make sure that you include the following as we cannot proceed without these:

- 1) A text file of the manuscript (doc, txt, rtf or tex), including the references, tables (including captions) and figure captions. Please remove any tracked changes from the text before submission. PDF files are not an accepted format for the "Main Document".
- 2) A separate electronic file of each figure (tif, eps or print-quality pdf preferred). The format should be produced directly from original creation package, or original software format.
- 3) Electronic Supplementary Material (ESM): all supplementary materials accompanying an accepted article will be treated as in their final form. Note that the Royal Society will not edit or typeset supplementary material and it will be hosted as provided. Please ensure that the

supplementary material includes the paper details where possible (authors, article title, journal name). Supplementary files will be published alongside the paper on the journal website and posted on the online figshare repository (<https://figshare.com>). The heading and legend provided for each supplementary file during the submission process will be used to create the figshare page, so please ensure these are accurate and informative so that your files can be found in searches. Files on figshare will be made available approximately one week before the accompanying article so that the supplementary material can be attributed a unique DOI. Alternatively you may upload a zip folder containing all source files for your manuscript as described above with a PDF as your "Main Document". This should be the full paper as it appears when compiled from the individual files supplied in the zip folder.

Article Funder

Please ensure you fill in the Article Funder question on page 2 to ensure the correct data is collected for FundRef (<http://www.crossref.org/fundref/>).

Media summary

Please ensure you include a short non-technical summary (up to 100 words) of the key findings/importance of your paper. This will be used for to promote your work and marketing purposes (e.g. press releases). The summary should be prepared using the following guidelines:

*Write simple English: this is intended for the general public. Please explain any essential technical terms in a short and simple manner.

*Describe (a) the study (b) its key findings and (c) its implications.

*State why this work is newsworthy, be concise and do not overstate (true 'breakthroughs' are a rarity).

*Ensure that you include valid contact details for the lead author (institutional address, email address, telephone number).

Cover images

We welcome submissions of images for possible use on the cover of Proceedings A. Images should be square in dimension and please ensure that you obtain all relevant copyright permissions before submitting the image to us. If you would like to submit an image for consideration please send your image to proceedingsa@royalsociety.org

Open Access

You are invited to opt for open access, our author pays publishing model. Payment of open access fees will enable your article to be made freely available via the Royal Society website as soon as it is ready for publication. For more information about open access please visit <https://royalsociety.org/journals/authors/open-access/>. The open access fee for this journal is £1700/\$2380/€2040 per article. VAT will be charged where applicable. Please note that if the corresponding author is at an institution that is part of a Read and Publishing deal you are required to select this option. See <https://royalsociety.org/journals/librarians/purchasing/read-and-publish/read-publish-agreements/> for further details.

Once again, thank you for submitting your manuscript to Proceedings A and I look forward to receiving your revision. If you have any questions at all, please do not hesitate to get in touch.

Best wishes
Raminder Shergill
proceedingsa@royalsociety.org
Proceedings A

Reviewer(s)' Comments to Author:

Referee: 1

Comments to the Author(s)

The authors present a detailed design and analysis of achromatic doublet lenses at hard X-ray wavelengths, in particular also considering the stretching effect of such lenses on ultrashort X-ray pulses. Here, the authors place emphasis on the thorough treatment of refraction through extended compound refractive lenses, which previously had not been done. Based on this analysis, the authors also present a few helpful examples.

The paper is well written, and the results are presented in a clear fashion. The application of this research lies in broadband and ultrafast hard X-ray science, a field which is rapidly developing. The present work has the potential to add to this development and may therefore become more relevant in the coming years. However, the paper is clearly written with this community in mind, and the overall situation of the field is not well introduced. This makes the context of the paper harder to understand for readers without in-depth knowledge of the state-of-the-art of attosecond free electron lasers. Furthermore, I have listed a few remarks and questions below. Assuming that the authors address these, I recommend publication in Proceedings of the Royal Society A.

- The achromatic optics of obvious interest for the focusing of attosecond pulses in the hard X-ray range. In my current understanding, this is an active field of research with frequent advances in the achievable pulse duration. It would be interesting if the authors can comment on the availability of such pulses and thereby provide a context in which these optics would be of value. Would these optics still provide a clear benefit for "longer" X-ray pulses, where the requirement on pulse stretching is less stringent?

- An obvious next lens design would be one where the diffractive lens is placed "inside" of the compound refractive lens. I realize that in this case, the mathematical description becomes more complex, but is it necessarily the case that this also leads to lower efficiency, as is claimed in the conclusion? Do the authors perhaps also foresee some benefit to this arrangement?

- In the current study, analysis was limited to the focusing of collimated X-ray beams. Could similar optics also be used to image and possibly (de)magnify a prior focus?

- The authors discuss that MLLs are often fabricated as cylindrical lenses. The consequences of such a fabrication approach are discussed, but I find this discussion a bit confusing. In the situation where two MLL's are crossed and stuck together, does this have any implications for the focusing and aberrations? The use of a pair of cylindric achromats is proposed as solution for when the MLL's cannot be brought in contact. Does this solution then have any implications for the resulting focus? I would expect that this may lead to ellipticity or even astigmatism in the focus.

- In the conclusion, the authors write about the described optical systems that their "focal length do vary considerably with wavelength", however also that the image position is constant. Considering the composite lens as a single element, this appears to be self-contradictory. Is this because not only the focus length but also the principle plane shifts with wavelength?

- One of the proposed applications of this lens is ptychography. There are of course many different implementations of this technique with varying requirements, but generally, it is thought that a structured probe beam is preferred above a diffraction-limited beam (e.g. Odstrčil, Opt. Express 27(10), 2019). The benefit of structured beams is even greater if one aims to reconstruct individual spectral components of a broadband spectrum. Is the pulse stretching of other, 'structured' focusing optics so significant that these achromatic doublets provide a clear advantage that really offsets the lack of structured focusing?

Some details:

- Attosecond pulses are routinely generated in the soft-X-ray and extreme ultraviolet regimes, however the presented solution will likely not work because of increased absorption. The authors might consider adding the distinction "hard X-ray" to the title to clarify this.

- The x-axis and it's labeling in Fig. 5 are not self-explanatory; do the authors mean something like $(\lambda - \lambda_0) / \lambda_0$?

- In table 1, several illuminating examples are given. For clarity, it would be beneficial to mention there also which materials were chosen for each example.
- The used symbols are generally chosen intuitively and are generally easy to remember. However, given the large number of different symbols, the authors might consider adding a list of symbols and their explanation.
- Page 3, line 53, 54 (“... and the consequent ...”): this sentence is hard to understand.

Referee: 2

Comments to the Author(s)

The authors present a theoretical analysis on the combination of diffractive and refractive lenses to achieve achromatic focusing in the X-ray regime. They thoroughly investigate several focusing solutions exploring different possible designs. They also show that with the appropriate separation between the refractive and diffractive lens, an apochromatic system can be built. The authors also demonstrate that such systems could be used to focus short X-ray pulse without distorting them in time by more than a few attoseconds, with very clear applications in temporal resolution investigations in X-ray free electron lasers. The manuscript is very well written and it presents the topic and findings in a very clear manner.

I recommend the publication of the manuscript after addressing the minor particular points of the following list:

- 1) The dispersive power, V , could already be defined in the introduction section in advance of section 2.
- 2) In page 4, line 13, is the chromatic condition missing a negative sign?
- 3) Figure 1 could be separated in two, splitting the diagrams for thin and thick lenses that actually belong to different sections of the manuscript.
- 4) Some of the labels in figure 1, such as F_i or $b(I)$ should be better clarified in the text or figure caption. The principal planes could also be displayed for the thin lenses for comparison with the thick lenses.
- 5) Typo in caption of figure 3, “Landscapes of the achromatic...”
- 6) Typo and space is missing in Page 19, line 29 “theCTRL ...”
- 7) In Figure 8, rays of two different colors could be used to display two different wavelengths.

Additional citations required (if applicable):

Board member

Board Member post-review comments:

Comments to Author(s):

I am happy for the paper to be accepted if the (minor) revisions from both referees are addressed.

Board Member pre-review comments:

Comments to Author(s):

Very high - comprehensive, sensible, research into the possibility of using an achromat (+ apochromat) lens doublet for X-ray focusing. Two main configurations discussed at length with accompanying ray-tracing.

Author's Response to Decision Letter for (RSPA-2021-0334.R0)

See Appendix A.

Decision letter (RSPA-2021-0334.R1)

15-Jun-2021

Dear Dr Chapman

I am pleased to inform you that your manuscript entitled "High-resolution achromatic X-ray optical systems for broad-band imaging and for focusing attosecond pulses" has been accepted in its final form for publication in Proceedings A.

Our Production Office will be in contact with you in due course. You can expect to receive a proof of your article soon. Please contact the office to let us know if you are likely to be away from e-mail in the near future. If you do not notify us and comments are not received within 5 days of sending the proof, we may publish the paper as it stands.

Under the terms of our licence to publish you may post the author generated postprint (ie. your accepted version not the final typeset version) of your manuscript at any time and this can be made freely available. Postprints can be deposited on a personal or institutional website, or a recognised server/repository. Please note however, that the reporting of postprints is subject to a media embargo, and that the status the manuscript should be made clear. Upon publication of the definitive version on the publisher's site, full details and a link should be added.

You can cite the article in advance of publication using its DOI. The DOI will take the form: 10.1098/rspa.XXXX.YYYY, where XXXX and YYYY are the last 8 digits of your manuscript number (eg. if your manuscript number is RSPA-2017-1234 the DOI would be 10.1098/rspa.2017.1234).

For tips on promoting your accepted paper see our blog post:
<https://royalsociety.org/blog/2020/07/promoting-your-latest-paper-and-tracking-your-results/>

On behalf of the Editor of Proceedings A, we look forward to your continued contributions to the Journal.

Sincerely,
Raminder Shergill
proceedingsa@royalsociety.org

Appendix A

Authors' response for Manuscript ID RSPA-2021-0334

We thank both reviewers for their careful reading and consideration of our paper. Our responses to the remarks and suggestions of the reviewers are interleaved below in blue. We also updated the funding statement and corrected an error in the penultimate paragraph of Section 5. Changes to the text can be seen in the red-line version.

Referee: 1

Comments to the Author(s)

The authors present a detailed design and analysis of achromatic doublet lenses at hard X-ray wavelengths, in particular also considering the stretching effect of such lenses on ultrashort X-ray pulses. Here, the authors place emphasis on the thorough treatment of refraction through extended compound refractive lenses, which previously had not been done. Based on this analysis, the authors also present a few helpful examples.

The paper is well written, and the results are presented in a clear fashion. The application of this research lies in broadband and ultrafast hard X-ray science, a field which is rapidly developing. The present work has the potential to add to this development and may therefore become more relevant in the coming years. However, the paper is clearly written with this community in mind, and the overall situation of the field is not well introduced. This makes the context of the paper harder to understand for readers without in-depth knowledge of the state-of-the-art of attosecond free electron lasers. Furthermore, I have listed a few remarks and questions below. Assuming that the authors address these, I recommend publication in Proceedings of the Royal Society A.

- The achromatic optics of obvious interest for the focusing of attosecond pulses in the hard X-ray range. In my current understanding, this is an active field of research with frequent advances in the achievable pulse duration. It would be interesting if the authors can comment on the availability of such pulses and thereby provide a context in which these optics would be of value. Would these optics still provide a clear benefit for "longer" X-ray pulses, where the requirement on pulse stretching is less stringent?

We have added some additional introductory material to introduce X-ray sources of attosecond pulses and that are used in broad band applications. As pointed out already in the introduction, there certainly are needs for achromats even when the pulse duration is not of interest, since they increase the throughput of the optical system, reducing exposure times and making better use of the available flux. We also added an additional word, "for" in the title to make it clearer that broadband and attosecond sources can be considered as distinct cases and that broad-band does not necessarily imply short pulses.

- An obvious next lens design would be one where the diffractive lens is placed "inside" of the compound refractive lens. I realize that in this case, the mathematical description becomes more complex, but is it necessarily the case that this also leads to lower efficiency, as is claimed in the conclusion? Do the authors perhaps also foresee some benefit to this arrangement?

We did take a look at these systems and find that the total length of the refractive lenses is higher when they surround a diffractive lens. So these do seem to lead to lower efficiency. There might be an advantage to use the additional degrees of freedom to correct for chromatic magnification or to further increase the bandwidth.

- In the current study, analysis was limited to the focusing of collimated X-ray beams. Could similar optics also be used to image and possibly (de)magnify a prior focus?

Yes, similar optics can be used for a finite source distance. If one is demagnifying a source, then the source distance is usually much larger than the focal length, and the analysis will be only slightly different to what is shown here. However, the analysis will be more complex without really giving more insight. For a magnifying geometry, one only needs to turn the system around or to think of the image as the source. The system that is most far away from the analysis presented here is 1:1 imaging. That would require some re-analysis, but the methodology to do so is all laid

out in the paper. We inserted a sentence in Sec. 1 (b) to explain that the collimating geometry covers most situations.

- The authors discuss that MLLs are often fabricated as cylindrical lenses. The consequences of such a fabrication approach are discussed, but I find this discussion a bit confusing. In the situation where two MLL's are crossed and stuck together, does this have any implications for the focusing and aberrations? The use of a pair of cylindric achromats is proposed as solution for when the MLL's cannot be brought in contact. Does this solution then have any implications for the resulting focus? I would expect that this may lead to ellipticity or even astigmatism in the focus.

The main consequence of using two crossed cylindrical lenses is anamorphism—that is, the magnification is different in the two directions. Astigmatism only arises if the two lenses do not focus to the same plane. In the situation of focusing a small source (on axis), this would have no effect. The lenses could have differing numerical apertures, which would have the same effect as a single lens with a rectangular pupil. We added reference to a recent analysis of the aberrations of crossed MLLs.

- In the conclusion, the authors write about the described optical systems that their “focal length do vary considerably with wavelength”, however also that the image position is constant. Considering the composite lens as a single element, this appears to be self-contradictory. Is this because not only the focus length but also the principle plane shifts with wavelength?

Yes, exactly. Only an achromatic doublet produces an image at the back focal plane. It does sound paradoxical that the focal length is different to the image distance, but this is well known in the design of achromatic systems and we were careful to introduce this earlier in the text. We added a reference in the conclusion back to the section where we introduced it.

- One of the proposed applications of this lens is ptychography. There are of course many different implementations of this technique with varying requirements, but generally, it is thought that a structured probe beam is preferred above a diffraction-limited beam (e.g. Odstrčil, Opt. Express 27(10), 2019). The benefit of structured beams is even greater if one aims to reconstruct individual spectral components of a broadband spectrum. Is the pulse stretching of other, ‘structured’ focusing optics so significant that these achromatic doublets provide a clear advantage that really offsets the lack of structured focusing?

This is an interesting question, and might deserve a whole other paper to properly address! We could imagine that it would be beneficial for ptychography of 3D objects to have a fair degree of dispersion, to create diversity along the beam axis for the same reason as for creating transverse structure. And one might prefer mixing the wavelengths in depth, instead of having an almost-linear variation as you would with a single diffractive or refractive lens. The achromat might be a good starting point from which to create an optimal structured probe for ptychography.

Some details:

- Attosecond pulses are routinely generated in the soft-X-ray and extreme ultraviolet regimes, however the presented solution will likely not work because of increased absorption. The authors might consider adding the distinction “hard X-ray” to the title to clarify this.

It is true that the systems made of solid materials would be too absorbing for soft X-rays, but it is interesting that thick refractive lenses using gas and plasmas will work for soft X-rays. So the concepts here will pertain to those cases and we would like to leave the title without the extra qualification. We do use “hard X-ray” in the abstract.

- The x-axis and it's labeling in Fig. 5 are not self-explanatory; do the authors mean something like $(\lambda - \lambda_0) / \lambda_0$?

Yes, indeed. We defined $\Delta \lambda$ in the text as $\lambda - \lambda_0$, so we should have used the label $\Delta \lambda / \lambda_0$. To avoid confusion, we adopted the suggestion of the reviewer.

- In table 1, several illuminating examples are given. For clarity, it would be beneficial to mention there also which materials were chosen for each example.

All these cases were all calculated for diamond, which we added to the caption of the table.

- The used symbols are generally chosen intuitively and are generally easy to remember. However, given the large number of different symbols, the authors might consider adding a list of symbols and their explanation.

This is a good idea. We have added a glossary table.

- Page 3, line 53, 54 (“... and the consequent ...”): this sentence is hard to understand.

We have rewritten the sentence.

Referee: 2

Comments to the Author(s)

The authors present a theoretical analysis on the combination of diffractive and refractive lenses to achieve achromatic focusing in the X-ray regime. They thoroughly investigate several focusing solutions exploring different possible designs. They also show that with the appropriate separation between the refractive and diffractive lens, an apochromatic system can be built. The authors also demonstrate that such systems could be used to focus short X-ray pulse without distorting them in time by more than a few attoseconds, with very clear applications in temporal resolution investigations in X-ray free electron lasers. The manuscript is very well written and it presents the topic and findings in a very clear manner.

I recommend the publication of the manuscript after addressing the minor particular points of the following list:

- 1) The dispersive power, V , could already be defined in the introduction section in advance of section 2.

This is a good suggestion, we have made this change.

- 2) In page 4, line 13, is the chromatic condition missing a negative sign?

Yes, thank you for spotting this. We have made the correction.

- 3) Figure 1 could be separated in two, splitting the diagrams for thin and thick lenses that actually belong to different sections of the manuscript.

We prefer to keep these together so that the reader can more easily see the connections and similarities between them.

- 4) Some of the labels in figure 1, such as F_i or $b(l)$ should be better clarified in the text or figure caption. The principal planes could also be displayed for the thin lenses for comparison with the thick lenses.

We have added explanations for the labels in the caption. The principal planes for the thin lenses overlap each other and are in the plane of the lens. They are in fact shown for the thin refractive lenses in (a) and (b). We changed the colour of the dashed lines from black to blue to make them consistent with the lines used for the principal planes of the thick lenses.

- 5) Typo in caption of figure 3, “Landscapes of the achromatic...”

We are not sure what the typo is. “Landscape” is correctly spelled, but perhaps can be misinterpreted. We changed the figure caption to remove this possibility.

- 6) Typo and space is missing in Page 19, line 29 “theCTRL ...”

We corrected this

7) In Figure 8, rays of two different colors could be used to display two different wavelengths.

For the achromat shown in Fig. 8, rays of two wavelengths that differ from their mean by 10 % do not disperse very much within the refractive lens and the two colours almost overlap and cannot be distinguished. when this is tried. The main aim of this figure is to show the condition where the MLL operates in an almost 1:1 geometry, and adding another ray colour is a distraction.